## [Transparent Peer Review file · Nature Communications]

Synthesis of 2D amorphous carbons via energy-autonomous carbonization of polyaniline upon decomposition of HClO₄

Corresponding Author: Professor Bastian Etzold

Version 0:

Reviewer comments:

Reviewer #1

(Remarks to the Author)

This is an interesting work well worth publishing in this journal. The authors utilize the energy released from a chemical reaction to achieve the transformation of polyaniline into amorphous carbon. Impressively, the amorphous carbon is well characterized using advanced techniques such as synchrotron radiation, neutron pair distribution function analysis, and high-resolution TEM. The authors also provide theoretical insights into the reaction mechanism. However, many important experimental details are not clearly described. Some experiments lack quantitative analysis. While the authors demonstrate that this is an effective method for producing amorphous carbon, there is limited discussion on how key variables—such as precursor amount, and moisture content in PANI, the doped HClO₄ content—affect the morphology and yield of the product. This manuscript could be suitable for publication after revision. Below are some questions and suggestions for the authors to address.

1. While the occurrence of a chemical reaction is theoretically governed by its activation energy barrier, it is unclear why the reaction temperature decreases with an increasing heating rate (e.g., 86 °C at 30 °C/min vs. 121 °C at 15 °C/min). Could the authors clarify the underlying mechanism? Furthermore, considering the heating rates provided, the system would require several minutes to reach the temperature necessary for initiating the energy-autonomous reaction. In contrast, Figure 1a illustrates only 0.4 seconds of the process, which may unintentionally give a misleading impression.
2. The reported 90% mass loss implies a maximum product yield of only 10%, which raises concerns about the efficiency of the process. Could the authors elaborate on the reason for this low yield and suggest possible strategies to improve it? Moreover, considering the system is open and the reaction appears explosive, has the formation of toxic gases such as hydrogen cyanide (HCN) been considered? What are the main gaseous products generated during the explosive reaction?
3. The authors demonstrated the structure of monolayer amorphous carbon using TEM. Does this mean that the synthesis method can directly produce monolayer amorphous carbon? If so, why does the short-duration, high-temperature reaction result in monolayer amorphous carbon rather than aggregated carbon structures? If not, please provide a detailed explanation of how the monolayer amorphous carbon was obtained.
4. In the Carbon K-edge XANES spectrum, the sp² characteristic peak near 285 eV appears to be poorly resolved. Although it is marked in Figure 2i, the corresponding feature is not clearly discernible, which is totally different from the sharp peak that appears in Figure 2h. Could the authors clarify this observation?
5. Although the reported temperature of the heated sample is only 86 or 120 °C, the energy-autonomous carbonization reaction generated visible flame or spark (Figure 1a), suggesting a much higher local temperature. Could the authors clarify the actual reaction temperature?
6. This work demonstrates that HClO₄-doped PANI undergoes a violent reaction upon heating up to 86 or 120 °C. However, many aspects lack quantitative discussion. For instance, (1) PANI is synthesized via interfacial polymerization of aniline with ammonium persulfate in the presence of HClO₄, and the presence of HClO₄ appears to be critical for initiating the energy-autonomous carbonization. Therefore, does the doping level of HClO₄ in PANI affect the reaction temperature and further affect the formation of the resulting amorphous carbon? How can the HClO₄ doping level be controlled? (2) The synthesized

PANI fibers were centrifuged from the aqueous phase and dried in air at 60 °C for 22 hours. However, they still exhibited a water content of 60–160% when transferred to the three-neck flask for reaction. Could the authors clarify why such high water retention occurred, and how the water content can be effectively controlled? The manuscript also mentions that the successful execution of the popping reaction is also highly dependent on the presence of a specific quantity of water within the PANI matrix. A quantitative analysis is needed to evaluate whether different levels of water content influence the occurrence of the reaction and the formation of amorphous carbon.

Reviewer #2

(Remarks to the Author)

The subject of this manuscript is interesting and aligns well with the scope of Nature Communications. The work is well-structured, and the experimental results are presented clearly. However, the authors should address the following points and revise the manuscript accordingly. Therefore I recommend major revisions before publication.

1. Introduction – This section requires improvement.

The authors should discuss current trends in the ultrafast and spontaneous synthesis of carbon nanostructures. In this context, numerous recent studies have reported innovative approaches for producing amorphous carbon nanostructures (e.g., nanosheets, carbon dots, porous carbons), including those synthesized in flame-based environments via hypergolic reactions. These reactions, which employ an organic fuel and an oxidizer, are carried out under ambient conditions without the need for sophisticated apparatus. Interestingly, they share similarities in terms of reaction rapidity, popping and the audible sound generated during the process.

Please clarify how your synthesis differs from these methods (similarities etc). Relevant works on ultrafast carbon synthesis that should be cited include:

<https://pubs.acs.org/doi/full/10.1021/acsnano.4c10531>

<https://pubs.acs.org/doi/full/10.1021/acs.chemmater.4c02091>

<https://www.mdpi.com/1420-3049/26/6/1595>

2. Terminology – Consider replacing the term explosion with ignition, as the former may convey a negative connotation to readers.

3. Reaction Yield and Atmosphere – What is the yield of the reaction? Possibly you face lower yield than expected due to the ignition and the popping effect. Do you anticipate any differences if the reaction is conducted under an inert atmosphere? Is the reaction scalable?

4. Fuel Cell Performance – Fuel cell measurements have become a standard method for evaluating performance under realistic operating conditions. Consider including such data.

5. Language and Clarity – The manuscript would benefit from further refinement of language and sentence structure to improve clarity and readability.

6. XPS Fitting Accuracy – Verify the accuracy of the XPS fitting in the Supplementary Information. In particular, in Figures S17–S19, the FWHM of the deconvoluted peak attributed to oxidized nitrogen appears larger compared to the other components.

Version 1:

Reviewer comments:

Reviewer #1

(Remarks to the Author)

The authors have satisfactorily addressed all comments from the previous review. The revised manuscript includes extensive additional experiments and comprehensive characterization, which provide strong and convincing support for the conclusions. The responses are clear and technically sound, and the new data substantially improve the rigor and completeness of the study. No additional revisions are required. I support acceptance of this manuscript for publication in Nature Communications.

Reviewer #2

(Remarks to the Author)

The authors have carefully addressed all the comments and requests raised during the review process, resulting in a significantly improved manuscript. I thank the authors for their thorough and thoughtful revisions.

The work is now suitable for publication.

Reviewer #1:

This is an interesting work well worth publishing in this journal. The authors utilize the energy released from a chemical reaction to achieve the transformation of polyaniline into amorphous carbon. Impressively, the amorphous carbon is well characterized using advanced techniques such as synchrotron radiation, neutron pair distribution function analysis, and high-resolution TEM. The authors also provide theoretical insights into the reaction mechanism. However, many important experimental details are not clearly described. Some experiments lack quantitative analysis. While the authors demonstrate that this is an effective method for producing amorphous carbon, there is limited discussion on how key variables—such as precursor amount, and moisture content in PANI, the doped HClO₄ content—affect the morphology and yield of the product. This manuscript could be suitable for publication after revision. Below are some questions and suggestions for the authors to address.

We sincerely thank the reviewer for his/her positive assessment of our work, in particular, for the insightful comments and questions, which have been invaluable in helping us improve the manuscript. In response, we conducted new experiments to systematically investigate the key factors identified by the reviewer: precursor amount, moisture content, heating rate, HClO₄ content, and HClO₄ type. This work focused on their effects on the reaction initiation temperature, product morphology, yield, and composition. The revised manuscript and supplementary information now include these experimental details and quantitative analyses. The new results and discussion, incorporated into the main text, have significantly deepened our understanding of the popping process mechanism. Below, we provide a point-by-point response to the reviewer's comments and questions.

Question 1.

While the occurrence of a chemical reaction is theoretically governed by its activation energy barrier, it is unclear why the reaction temperature decreases with an increasing heating rate (e.g., 86 °C at 30 °C/min vs. 121 °C at 15 °C/min). Could the authors clarify the underlying

mechanism? Furthermore, considering the heating rates provided, the system would require several minutes to reach the temperature necessary for initiating the energy-autonomous reaction. In contrast, Figure 1a illustrates only 0.4 seconds of the process, which may unintentionally give a misleading impression.

Response: We thank the reviewer for this insightful question and the opportunity to clarify the underlying mechanism. We agree with the reviewer that the reaction is governed by an activation barrier. The observed decrease in initiation temperature with increasing heating rate indicates that heat accumulation is critical for the popping reaction. This phenomenon can be explained by considering the system's thermal dynamics. The decomposition of HClO_4 , which triggers the overall popping reaction, requires a critical heat accumulation within the precursor. At a faster heating rate, heat is supplied to the system more rapidly than it can dissipate, leading to efficient heat accumulation. This localized temperature rise allows the system to reach the activation energy for HClO_4 decomposition at a lower measured temperature. Conversely, at a slower heating rate, heat loss to the surroundings is more significant, requiring a higher temperature to achieve the same critical internal heat accumulation to initiate the reaction. This effect is further modulated by the precursor's content and water content, which influence its overall heat capacity and thermal conductivity. The discussion has been added in the revised manuscript (Page 13):

While the reaction initiation temperature and overall POP-C yield are governed by external factors (e.g., PANI packing density and amount, heating rate, water content) as detailed in Supplementary Note 4, the successful formation of the desired 2D nanosheet morphology critically requires a high free HClO_4 content (mass content >30%), which ensures sufficient gas volume and exothermicity for complete carbonization and exfoliation.

And more detailed discussion is presented in the Supplementary Information on Page 38-42:

3. Influence of the heating rate

The popping reaction was conducted using an IKA RCT basic heating mantle with a metal bath. The heating rate was controlled by setting a target temperature, with higher target temperatures

resulting in faster heating rates (Supplementary Fig. 39). For instance, setting the target temperature to 300 °C and 200 °C yielded heating rates of 22 °C min⁻¹ (in the range of 85 to 150 °C) and 15 °C min⁻¹ (in the range of 100 to 150 °C), respectively. To achieve a slower heating rate, a stepped protocol was employed: the target temperature was initially set to 100 °C, then increased to 150 °C upon reaching 80 °C, to 180 °C upon reaching 130 °C, and finally to 210 °C upon reaching 160 °C. This stepped protocol lead to a heating rate of stepped 6 °C min⁻¹ in the range of 110 to 160 °C.

Supplementary Fig. 39 Heating rates for the IKA RCT basic heating mantle. The heating rate was calculated from a linear fit of the temperature vs time curve. The fitting range was selected based on the typical reaction temperature range: 85-150 °C for a rate of 6 °C min⁻¹, 100-150 °C for 15 °C min⁻¹, and 110-160 °C for 22 °C min⁻¹.

The PANI used to investigate the influence of heating rate was synthesized using 1.0 M HClO₄ as the aqueous phase, followed by water washing and drying at 60 °C for 22 h. Its composition was quantified as 32% PANI base, 15% doped HClO₄, 33% free HClO₄, and 20% water. Video analysis of POP-C synthesis from 100 mg PANI (Supplementary Video 6, key frames in Supplementary Fig. 40) reveals that the reaction becomes more intense at higher heating rates. The flame observed at 22 °C min⁻¹ is significantly brighter than those at 15 and 6 °C min⁻¹, indicating a more violent reaction.

Supplementary Fig. 40 Synthesis of POP-C from PANI via rapid thermal popping at different heating rates: (a) $6\text{ }^{\circ}\text{C min}^{-1}$, (b) $15\text{ }^{\circ}\text{C min}^{-1}$, and (c) $22\text{ }^{\circ}\text{C min}^{-1}$. The temperature displays alternates between the thermocouple reading (the actual temperature) and the set-point temperature. The popping reaction initiates at the thermocouple reading (e.g., $127\text{ }^{\circ}\text{C}$ in panel b), not at the set-point value.

Both the reaction initiation temperature and the final POP-C yield are influenced by the heating rate. As shown in Supplementary Fig. 41a, the initiation temperature decreases from $139\text{ }^{\circ}\text{C}$ to $129\text{ }^{\circ}\text{C}$ and finally to $113\text{ }^{\circ}\text{C}$ as the heating rate is increased from 6 to 15 to $22\text{ }^{\circ}\text{C min}^{-1}$. This trend occurs because a higher heating rate facilitates more efficient heat accumulation, thereby lowering the initiation temperature of the reaction. The product yield decreases from 34% to 32% to 26% with increasing heating rates (Supplementary Fig. 41b). This inverse relationship between yield and heating rate is attributed to greater product loss during the more violent

reactions; a larger portion of the POP-C is expelled from the flask by the rapid gas generation at higher heating rates and cannot be collected. This explanation is consistent with the observed increase in reaction intensity (i.e., brighter flame) at higher heating rates, as shown in Supplementary Fig. 40. The lowest triggering temperature observed was 86 °C, using 200 mg of PANI precursor and a heating rate of ~ 30 °C min^{-1} . **However, combining a large precursor mass with a high heating rate can produce an excessively violent reaction. Therefore, this specific combination is not recommended due to safety concerns.**

Supplementary Fig. 41 Effect of heating rate on the popping reaction and resulting POP-C. (a) The dependence of reaction initiation temperature on PANI mass. (b) POP-C yield as a function of PANI mass. (c) CV curves and (d) Raman spectra of POP-C synthesized at different heating rates.

The ECSA of the POP-C samples, synthesized with different heating rates, was evaluated by

CV in 0.1 M KOH. The highly rectangular shape of the CV curves, maintained even at a high scan rate of 100 mV s^{-1} , indicates a well-interconnected, open pore structure that facilitates rapid ion transport (Supplementary Fig. 41c). The near-identical CV profiles across all samples suggest that the heating rate during synthesis has a negligible influence on the resulting surface area and porous architecture. The comparable Raman spectra of POP-C synthesized under different heating rates (Supplementary Fig. 41d) indicate that this parameter has no significant influence on the carbon hybridization, defect density, or structural disorder of the final product.

We agree with the reviewer and appreciate the opportunity to clarify this question. Fig. 1a is intended to illustrate the brief, energy-autonomous “popping” event ($\sim 0.4 \text{ s}$), not the preceding heating period. To make this distinction unequivocal, we have modified the discussion on Page 3 and 4 in the revised manuscript to explicitly state the total heating time required for initiation:

The entire process is simple, conducted in a round-bottom flask without complex instrumentation or post-treatment, and completes within 2-7 min under ambient conditions.

The obtained PANI is then loaded into a round bottom flask fitted with a thermometer, covered with quartz wool to permit outgassing (Fig. 1 and Supplementary Fig. 2). Subsequent mild heating of the sample for 2-7 min initiates the energy-autonomous carbonization process, which culminates in a brief ignition (lasting $\sim 0.4 \text{ s}$) and a violent release of gas.

Furthermore, we have added the total reaction time in the Supplementary Table 5 to provide a clear comparison with conventional pyrolysis methods (Page 63 in the Supplementary Information):

Supplementary Table 5 Comparison of the energy consumption.

Methods	Operation power (W)	Operation time ^a (min)
Conventional pyrolysis	1000-4000	150-350
Popping by mild heating	100-400	2-7
Popping by microwave	1000	0.5-1

^a The operation time includes both ramps and dwells segments

Question 2.

The reported 90% mass loss implies a maximum product yield of only 10%, which raises concerns about the efficiency of the process. Could the authors elaborate on the reason for this low yield and suggest possible strategies to improve it? Moreover, considering the system is open and the reaction appears explosive, has the formation of toxic gases such as hydrogen cyanide (HCN) been considered? What are the main gaseous products generated during the explosive reaction?

Response: We sincerely thank the reviewer for raising this critical point regarding process efficiency. The reviewer is correct that a 90% mass loss suggests a 10% yield, which would indeed be low. Our initial calculation was based on the total precursor mass, which included non-carbonizing components. Following the reviewer's insightful suggestion, we have conducted a systematic quantitative analysis of the precursor composition. A typical popping-reactive PANI precursor comprises 32% PANI emeraldine base, 15% doped HClO₄, 33% free HClO₄, and 20% water. Since only the PANI emeraldine base serves as the carbon precursor converted into the final product, the yield should be calculated relative to this component's mass. Using this corrected calculation method: for a 200 mg precursor producing 20 mg of POP-C, the yield is $20 \text{ mg} / (200 \text{ mg} \times 32\%) = 31\%$. To ensure reliability, yields were determined from at least three independent reactions. The average yield for a standard popping reaction (200 mg PANI precursor, heated at $15 \text{ }^\circ\text{C min}^{-1}$) was 28%. The primary cause of product loss is the physical expulsion of POP-C by the violent gas release during the popping reaction. Consequently, moderating the reaction intensity through precursor and reaction engineering is key to improving yield. For instance, decreasing the PANI precursor amount yields a less violent reaction while still achieving a POP-C yield of 40%. Similarly, increasing the water content in the precursor decrease the reaction intensity, reducing the force of gas expulsion. This mild reaction minimizes product loss, resulting in a higher collected yield of 36%. We have revised the manuscript (Page 15) accordingly to present this more accurate yield and have provided a detailed quantitative analysis of the precursor composition and yield calculation in the Supplementary Information (Page 30-34, 36-37, 44-45) to ensure full

transparency. We believe this corrected framework accurately reflects the carbon conversion efficiency of the popping process.

Page 15 in the revised manuscript:

Moreover, the yield of POP-C from the popping reaction ranges from 17% to 40% depending on reaction conditions, which is comparable to that of conventional pyrolysis (37%), as detailed in Supplementary Notes 4 and 5.

Page 30-34 in the Supplementary Information:

As detailed in the Experimental section and Supplementary Fig. 1, the PANI precursor for POP-C was synthesized via interfacial polymerization of aniline. The oxidative polymerization occurs at the interface, forming polyaniline, which subsequently migrates into the aqueous phase (1.0 M HClO₄). The resulting product, confirmed to be emeraldine salt doped with HClO₄, was separated from the aqueous phase by centrifugation, washed with water, and air-dried at 60 °C for 22 h. The obtained PANI was in a wet, sticky state, indicating the likely presence of multiple components: the HClO₄ doped PANI emeraldine salt itself, residual free HClO₄ from the excess acid in the aqueous phase, and water. To quantitatively analyze these compositions, we developed a straightforward method based on acid-base titration, as illustrated in Supplementary Fig. 33.

Supplementary Fig. 33 Quantification protocol for the composition of the PANI precursor, including polyaniline base, doped HClO₄, free HClO₄, and water contents.

First, a known mass of the PANI sample was dispersed in an excess of 0.1 M KOH to ensure complete neutralization of all HClO₄ species. This includes both the doped HClO₄ (which forms a salt with the polyaniline backbone) and the free HClO₄ (residual acid from synthesis). The chemical reaction, in which the emeraldine salt is converted to the un-doped emeraldine base, is illustrated in Supplementary Fig. 34. Following this reaction, the PANI suspension turns from green to dark blue, visually confirming the formation of the emeraldine base (Supplementary Fig. 35a). After approximately 6 h, the PANI emeraldine base precipitates. The supernatant, containing the unreacted KOH, was then transferred to a beaker and titrated with 0.1 M HClO₄ using phenolphthalein (PhPh) as an indicator (Supplementary Fig. 35a). The amount of HClO₄ consumed in the initial neutralization was determined by calculating the difference between the initial KOH and the amount remaining after reaction, as quantified by the titration.

Supplementary Fig. 34 Schematic of the quantitative analysis for free and doped HClO₄ in PANI. (a) The PANI emeraldine salt is reacted with KOH. Titration of consumed KOH quantifies total HClO₄ (free and doped HClO₄). (b) The PANI emeraldine salt is dispersed in water. Titration of the aqueous phase quantifies free HClO₄. Doped HClO₄ is calculated by difference.

Supplementary Fig. 35 Quantitative analysis of HClO₄ in PANI. (a) Measurement of total HClO₄ content. (b) Measurement of free HClO₄ content. The doped HClO₄ content is the difference of total and free HClO₄ content.

Second, a known mass of the PANI sample was dispersed in water to dissolve the free HClO₄ (Supplementary Fig. 34) The suspension remained green, characteristic of the PANI emeraldine salt (Supplementary Fig. 35b). After approximately 6 h, the PANI emeraldine salt precipitated. The supernatant, which contained the dissolved free HClO₄, was transferred to a beaker and titrated with 0.1 M KOH using PhPh as an indicator (Supplementary Fig. 35b). The amount of free HClO₄ was directly quantified from the titration results. The mass of doped HClO₄ was subsequently calculated by subtracting the free HClO₄ content from the total HClO₄ content.

Third, a control sample of polyaniline without water and free HClO₄, designated PANI-D, was prepared. PANI-D was synthesized via the same interfacial polymerization of aniline as the standard PANI. The critical difference lies in the washing procedure: after separation from the aqueous phase, the polymer was washed with acetone to remove all water and free HClO₄, followed by air-drying at 60 °C for 22 h. The free HClO₄ originates from the residual HClO₄ solution trapped within the standard PANI matrix. Acetone washing effectively removes both this residual acid and the associated water. Consequently, PANI-D is obtained as a green powder, in contrast to the standard PANI, which contains water and free HClO₄ and presents as a black, sticky solid (Supplementary Fig. 1). The mass of polyaniline without doped HClO₄ (PANI base) was subsequently calculated by subtracting the doped HClO₄ content from the PANI-D content.

Finally, the water content can be obtained by subtracting the doped HClO₄ content, free HClO₄ content and PANI base content from the PANI content.

To investigate how the heating rate, precursor mass, water content, HClO₄ content, and the type of HClO₄ within the precursor influence the popping reaction, several PANI precursor were prepared and the PANI emeraldine base, doped HClO₄, free HClO₄ and water content was quantified based on the protocol in Supplementary Fig. 33. The quantitative compositional analysis via titration was performed at least three times for each sample to ensure reproducibility and to generate error bars.

Page 36-37 in the Supplementary Information:

The yield of POP-C was calculated based on the mass of the PANI emeraldine base in the precursor. This is because only the emeraldine base component, excluding the doped HClO₄, free HClO₄ and water, is considered as the carbon precursor converted into the final carbonaceous product. For the PANI used in this series of experiments (comprising 32% PANI base, 15% doped HClO₄, 33% free HClO₄, and 20% water), the yield decreased from 40% to 32% to 28% with increasing precursor mass (Supplementary Fig. 38b). This trend is attributed to product loss during the violent reaction; a portion of the POP-C is expelled from the flask

by the rapid gas generation and cannot be collected. Since the reaction intensity diminishes with lower precursor mass (as shown in Supplementary Fig. 36), the relative product loss is reduced, leading to a higher relative yield at lower masses.

Page 44 in the Supplementary Information:

The product yield increases from 17% to 28% to 33% and finally to 36% with increasing water content (Supplementary Fig. 43b). This positive correlation is attributed to the less violent popping process observed at higher hydration levels. As the water content increases, the reaction intensity diminishes, resulting in less POP-C being expelled from the flask by rapid gas generation. The milder reaction thus minimizes product loss and leads to a higher collected yield.

A key consideration for scaling the popping reaction is managing its vigor. We have demonstrated that the reaction intensity can be effectively modulated by adjusting the water content of the precursor; higher water content reduces the intensity, albeit at the cost of a higher triggering temperature. *Therefore, precursors with elevated water content present a viable pathway for safer scale-up.*

We thank the reviewer for raising this crucial point regarding the safety and gaseous byproducts of the reaction. We agree that identifying potential toxic gases is essential for evaluating the process. Our analysis confirms that the primary gaseous products are CO₂, H₂O, and Cl₂, with no detectable formation of HCN. The violent gas generation is intrinsic to the mechanism, as the decomposition of HClO₄ provides both the energy for carbonization and the gaseous species for exfoliation.

All popping reactions were conducted in a fume hood. The formation of Cl₂ gas, a decomposition product of HClO₄, was confirmed during the reaction using a dedicated chlorine sensor placed within the fume hood. We collected the gaseous products and performed IR spectroscopy. The resulting spectrum shows clear signals for CO₂ and H₂O, but shows no IR absorption corresponding to the C≡N stretch (2100-2200 cm⁻¹), effectively ruling out HCN as a significant byproduct. Therefore, we conclude that Cl₂ is the main hazardous product and

must be managed in a fume, while the formation of HCN is highly unlikely. We have added this discussion and the supporting IR data to the Supplementary Information to ensure full transparency regarding the reaction's safety profile. An explicit safety warning has also been included, emphasizing that all procedures must be conducted in a fume hood coupled with chlorine sensor due to Cl₂ generation.

Page 56 in the Supplementary Information:

During the reaction, chlorine can be detected in the fume hood by using a chlorine gas sensor, which is due to the decomposition of HClO₄. The decomposition of HClO₄ can also generate H₂O and O₂:

In the presence of polyaniline, additional gases are formed due to oxidation at high local temperatures. IR spectroscopy confirms the release of CO₂ (Supplementary Fig. 51), resulting from the strong oxidative carbonization of the PANI backbone.

Supplementary Fig. 51 IR spectrum of the gaseous products released during the popping reaction.

Safety Warnings:

The decomposition of HClO₄ produces Cl₂ gas. All reactions must be conducted in a fume hood equipped with a chlorine gas sensor.

Page 14-15 in the revised manuscript:

The decomposition of perchlorate species also concomitantly releases a large volume of gases (e.g., Cl₂, O₂, CO₂, H₂O). The instantaneous release of energy drives rapid dehydrogenation and carbonization, with local temperature confirmed to exceed 1400 °C (Supplementary Note 4). Simultaneously, the violent gas evolution creates a critical internal pressure that physically exfoliates the carbonizing polymer matrix from the inside out, while the in-situ gas-bubble templating directly yields the characteristic 2D porous nanosheet morphology with an open, interconnected pore network (Supplementary Fig. 47). This synergy between intense localized heating and explosive gas expansion culminates in a thermal runaway event, observed as a rapid “pop”.

Question 3.

The authors demonstrated the structure of monolayer amorphous carbon using TEM. Does this mean that the synthesis method can directly produce monolayer amorphous carbon? If so, why does the short-duration, high-temperature reaction result in monolayer amorphous carbon rather than aggregated carbon structures? If not, please provide a detailed explanation of how the monolayer amorphous carbon was obtained.

Response: We thank the reviewer for this critical question regarding the formation mechanism of the monolayer structure. In short, yes, our synthesis method directly produces 2D amorphous carbon nanosheets in a single step. The short-duration, high-temperature reaction alone is insufficient; the key is the synergy between rapid carbonization and explosive gas exfoliation, which forcibly separates the layers before they can aggregate. The mechanism involves two concurrent processes: instantaneous carbonization and explosive exfoliation. The exothermic decomposition of energetic components (e.g., free HClO₄) generates intense local heat (>1400 °C), driving rapid dehydrogenation and the formation of sp² carbon from the PANI backbone. The same decomposition violently releases a large volume of gases (e.g., Cl₂, O₂, CO₂, H₂O). This creates a critical internal pressure that physically forces the carbonizing intermediate apart, culminating in the observable “pop”. This combined effect is what

simultaneously carbonizes and exfoliates the precursor, directly yielding the 2D nanosheet morphology. In contrast, conventional pyrolysis lacks this violent gas evolution, leading to a gradual process that retains the precursor's agglomerated morphology (Supplementary Fig. 31g-h). Our controlled experiments provide direct evidence for this mechanism. The formation of the 2D nanosheet morphology is directly correlated with the free HClO₄ content, which dictates the volume of gas generated. Precursors with low free HClO₄ content (< 20%) carbonize without full exfoliation, resulting in more aggregated structures. Only above a critical threshold does the gas pressure become sufficient to produce the characteristic 2D nanosheets (Supplementary Fig. 47). Raman spectroscopy further supports this. The increase of the I_D/I_G ratio with rising HClO₄ content correlates with the progression from partial to complete exfoliation, the latter yielding nanosheets with a high density of crystallite boundaries (Supplementary Fig. 46f). In conclusion, the popping transformation is defined not merely by rapid heating but by the combined effect of intense, localized heating and explosive in-situ gas generation. The gas pressure acts as a physical exfoliation agent, directly yielding 2D amorphous carbon and preventing aggregation. We have revised the relevant sections of the revised manuscript (Page 13 and 15) and Supplementary Information (Page 29-30, 50-53) to present this refined mechanism, substantiated by our new data.

Page 14-15 in the revised manuscript:

Based on these findings, we propose a synergistic mechanism for the energy-autonomous carbonization and exfoliation of PANI. As illustrated in Supplementary Fig. 53, the transformation requires a PANI emeraldine salt containing a critical amount of free HClO₄. Upon heating, HClO₄ decomposes, initiating a sequence of events. First, the decomposition generates potent radical species and oxidants, such as ClO₂, ClO, and O₂⁴¹, triggering a violent, highly exothermic oxidation of PANI, which has long been recognized as an effective radical scavenger/heat insulator, and can be used to desensitize nanothermites⁴²⁻⁴⁴. The decomposition of perchlorate species also concomitantly releases a large volume of gases (e.g., Cl₂, O₂, CO₂, H₂O). The instantaneous release of energy drives rapid dehydrogenation and carbonization,

with local temperature confirmed to exceed 1400 °C (Supplementary Note 4). Simultaneously, the violent gas evolution creates a critical internal pressure that physically exfoliates the carbonizing polymer matrix from the inside out, while the in-situ gas-bubble templating directly yields the characteristic 2D porous nanosheet morphology with an open, interconnected pore network (Supplementary Fig. 47). This synergy between intense localized heating and explosive gas expansion culminates in a thermal runaway event, observed as a rapid “pop”. While the reaction initiation temperature and overall POP-C yield are governed by external factors (e.g., PANI packing density and amount, heating rate, water content) as detailed in Supplementary Note 4, the successful formation of the desired 2D nanosheet morphology critically requires a high free HClO₄ content (mass content >30%), which ensures sufficient gas volume and exothermicity for complete carbonization and exfoliation.

Page 15 in the revised manuscript:

In stark contrast, the popping reaction described herein is characterized by the decomposition of energetic components in the PANI precursor, triggering rapid carbonization in less than one second. Concurrently, vigorous outgassing promotes the formation of 2D carbon nanosheet. The popping process exhibits strong potential for scale-up due to its speed, low external energy demand, and tunable reaction intensity, which can be moderated via precursor water content, as detailed in Supplementary Notes 4. Moreover, the yield of POP-C from the popping reaction ranges from 17% to 40% depending on reaction conditions, which is comparable to that of conventional pyrolysis (37%), as detailed in Supplementary Notes 4 and 5.

Page 29-30 in the Supplementary Information:

While PANI-D maintains a similar fibrous morphology to standard PANI (Supplementary Fig. 32e), it does not undergo the popping reaction. Heating PANI-D to 150 °C results in a dense material (PANI-D150) that retains its short fibrous structure (Supplementary Fig. 32f). Subsequent high-temperature pyrolysis (800 °C) of PANI-D yields a dense, agglomerated black powder (PANI-D800, Supplementary Fig. 31g-h), starkly contrasting with the fluffy, nanosheet

morphology of POP-C obtained from the popping reaction (Supplementary Fig. 32b-d). These findings highlight the crucial factors for the low-temperature popping reaction and raise central mechanistic questions: Why is HClO₄-doped PANI uniquely capable of this transformation? What specific components are removed by the acetone washing process that deactivate the popping reaction?

Supplementary Fig. 32 SEM images of (a) PANI, (b) POP-C and (c) POP-C800. (d) Volume comparison of 200 mg PANI, 20 mg POP-C and 20 mg POP-C800. SEM images of (f) PANI-D, (g) PANI-D150 and (h) PANI-D800. (e) Volume comparison of 200 mg PANI-D, 160 mg PANI-D150 and 50 mg PANI-D800.

Supplementary Fig. 46 Effect of HClO₄ content in PANI precursor on the popping reaction and resulting POP-C. (a) The reaction initiation temperature as a function of the free HClO₄ content. (b) The final yield of POP-C as a function of free HClO₄ content. (c) CV curves and (d) Raman spectra of POP-C synthesized from PANI precursors with varying free HClO₄ content.

Safety Warnings:

Polyaniline (PANI) composites containing elevated levels of HClO₄, especially free acid, are highly reactive and may combust spontaneously. To mitigate this risk, the preparation of precursors with total HClO₄ content >60% (free HClO₄ >47%) is strongly discouraged!

The ECSA of POP-C samples synthesized from PANI with different HClO₄ contents was evaluated by CV in 0.1 M KOH. POP-C derived from PANI with high free HClO₄ content (33%

and 47%) exhibits highly rectangular CV curves, even at a high scan rate of 100 mV s^{-1} , indicating a well-interconnected, open pore structure that facilitates rapid ion transport (Supplementary Fig. 46c). The comparable curves for these two samples suggest they possess similar ECSA values. POP-C from a medium free HClO_4 content (21%) also shows a rectangular shape but with a reduced double-layer current, indicating good conductivity yet a smaller ECSA. In contrast, POP-C from PANI with lower HClO_4 content (12% and 17%) shows a significantly smaller double-layer current and distorted, “leaf-like” CV profiles. This shape suggests high electrical resistance, mass transport limitations, a much smaller surface area, and a less developed pore structure. These inferior electrochemical properties are a direct result of incomplete exfoliation, caused by the smaller amount of gas generation associated with insufficient HClO_4 content.

Raman spectroscopy of the popping products (Supplementary Fig. 46d) confirms the formation of carbonaceous materials rather than PANI. With the increase of HClO_4 content from 12% to 17% to 21%, the I_D/I_G value increases from 0.85 to 0.87 to 0.95, while when the HClO_4 content increases from 21% to 33% to 47%, the I_D/I_G stays at the same level (0.95, 1.00 and 0.98). We propose that this trend is governed by the extent of exfoliation during the popping reaction. This reaction induces rapid dehydration of the polyaniline precursor, promoting the graphitization of its benzene rings into sp^2 carbon. Higher HClO_4 content elevates the reaction temperature and gas evolution, facilitating the formation and growth of sheet-like carbon. For precursors with low free HClO_4 content (12% and 17%), carbonization occurs without full exfoliation, resulting in a less disordered structure and a lower I_D/I_G ratio. In contrast, for precursors with high HClO_4 content (33% and 47%), carbonization is accompanied by complete exfoliation, yielding sheet-like nano-carbons with an abundance of crystallite boundaries, which is consistent with the observed higher I_D/I_G ratios³⁵.

Further TEM characterization of the POP-C samples revealed a clear morphological evolution directly correlated with the HClO_4 content (Fig. 5d-h). The POP-C derived from a precursor with 12% free HClO_4 content exhibits an agglomerated morphology (Supplementary Fig. 5d), indicating that the heat generated was sufficient for carbonization but the generated gas is

insufficient for exfoliation. As the HClO_4 content increased to 17%, the resulting POP-C began to show the presence of nanosheets alongside agglomerates (Fig. 5e), signifying the onset of partial exfoliation. At 21% HClO_4 , the product displayed an interconnected, thin-sheet structure, pointing to a more advanced exfoliation process (Fig. 5f). Finally, precursors with high HClO_4 content (33% and 47%) yielded POP-C consisting entirely of well-defined, 2D interconnected nanosheets (Fig. 5g-h). This systematic progression confirms that the violent gas evolution during the popping reaction is the critical factor for exfoliation. These observations provide direct visual evidence for the proposed formation mechanism: the popping reaction simultaneously drives rapid carbonization through intense local heating and forcibly exfoliates the intermediate through internal gas pressure, dynamically separating the polymer matrix into large-area, porous 2D nanosheets. This process also creates an overall porous foam morphology (Supplementary Fig. 47), resulting from the synergy between explosive exfoliation and in-situ gas-bubble templating. Consequently, the generation of gas (directly governed by the HClO_4 content in the precursor) is the fundamental driver of the unique 2D nanosheet architecture.

Supplementary Fig. 47 TEM image of Ni-POP-C, showing evidence of gas-bubble templating.

Question 4.

In the Carbon K-edge XANES spectrum, the sp^2 characteristic peak near 285 eV appears to be poorly resolved. Although it is marked in Figure 2i, the corresponding feature is not clearly discernible, which is totally different from the sharp peak that appears in Figure 2h. Could the authors clarify this observation?

Response: We appreciate this reviewer for this insightful question. We agree that the C-C π^* peak at ~285 eV appears sharper in the EELS spectrum compared to the broader feature observed in the XANES spectrum. The difference in the appearance of the ~285 eV feature arises from the fundamental distinction between the two techniques and the chemical heterogeneity of POP-C, rather than from the absence of sp^2 carbon. Specifically, the EELS data were acquired using a TEM instrumentation. As shown in Figure 2b, the POP-C material consists of thin, wrinkled 2D nanosheets. EELS is a transmission technique with high spatial resolution (sub-nm scale). When acquiring EELS, the beam is typically focused on a thin, clean region of the carbon nanosheet to avoid multiple scattering. Consequently, the EELS spectrum in Figure 2h primarily reflects the intrinsic “skeleton” of the carbon network, dominated by the sp^2 conjugated honeycomb lattice, resulting in a distinct and sharp π^* peak at 285 eV. While in contrast, the XANES spectrum (Figure 2i) is collected in TEY mode with a much larger beam footprint. It represents an ensemble average over many nanosheets, including edges, defects, and regions with varying orientation. This averaging naturally broadens spectral features. Moreover, POP-C contains substantial N and O dopants, as confirmed by XPS. Their associated π^* resonances (C–N, C–O) appear around 286–289 eV, overlapping with the C=C π^* transition. These contributions are prominent in the macroscopic XANES signal but much less evident in the localized EELS measurement, which may also contribute to a less resolved 285 eV feature in XANES (Zhong et al., *Carbon*, 2012, 50, 335–338). In summary, the sharp EELS peak reflects the well-defined sp^2 domains at the nanoscale, whereas the broader XANES feature reflects ensemble averaging over diverse chemical and structural environments. We have revised the revised manuscript (Page 7) to clarify that the broadening in XANES arises from the coexistence of graphitic regions and heteroatom-rich or disordered sites that are not equally

represented in the local EELS measurement.

Page 7 in the revised manuscript:

A distinction between the C K-edge EELS (Fig. 2h) and XANES (Fig. 2i) spectra is evident in the sharpness of the sp^2 C=C π^* feature near 285 eV. EELS reveals a well-defined π^* peak, whereas XANES shows a broadened signal. This difference arises from the distinct probing scales and averaging effects of these two techniques. TEM-EELS probes local, well-ordered sp^2 domains within individual nanosheets. In contrast, XANES provides a macroscopic average, where the sp^2 C=C π^* transition (~ 285.5 eV) overlaps with contributions from abundant C–N and C–O groups (~ 288.3 eV) in disordered regions. Therefore, the sharp EELS signal confirms the existence of well-developed sp^2 networks at the nanoscale, while the broader XANES signal reflects the chemically and structurally diverse, functionalized nature of the material on a macroscopic scale.

Question 5.

Although the reported temperature of the heated sample is only 86 or 120 °C, the energy-autonomous carbonization reaction generated visible flame or spark (Figure 1a), suggesting a much higher local temperature. Could the authors clarify the actual reaction temperature?

Response: We thank the reviewer for this excellent question. The reviewer is correct to note that the reported temperatures of 86 °C or 120 °C represent the triggering temperature for the reaction, not the actual temperature during the energy-autonomous event. Directly measuring the instantaneous reaction temperature is challenging due to the extremely short timescale (~ 0.4 s). However, we have obtained a reliable estimate for the peak temperature through an observation: when a K-type thermocouple was embedded in the packed PANI precursor, it melted during the popping reaction. Since the melting point of the standard nickel-chromium/nickel-alumel (K-type) thermocouple alloy is approximately 1400 °C, this observation provides direct evidence that the local temperature exceeds this value. We have incorporated this finding and its interpretation into both the revised manuscript and the

Supplementary Information to clarify the intense exothermic nature of the process.

Page 4 in the revised manuscript:

The instantaneous local reaction temperature exceeds 1400 °C, as confirmed by the melting of a K-type thermocouple (Supplementary Fig. 3).

Page 14 in the revised manuscript:

The instantaneous release of energy drives rapid dehydrogenation and carbonization, with local temperature confirmed to exceed 1400 °C (Supplementary Note 4).

Page 3-4 in the Supplementary Information:

The popping reaction is highly exothermic, generating substantial heat instantaneously. Although triggered at a moderate temperature, the local reaction temperature exceeds 1400 °C, as confirmed by the melting of a K-type thermocouple. For this reason, the thermocouple was placed adjacent to (rather than embedded within) the packed PANI precursor to avoid direct damage.

Supplementary Fig. 3 The melted K-type thermocouple after popping reaction.

Question 6.

This work demonstrates that HClO₄-doped PANI undergoes a violent reaction upon heating up to 86 or 120 °C. However, many aspects lack quantitative discussion. For instance, (1) PANI is synthesized via interfacial polymerization of aniline with ammonium persulfate in the presence of HClO₄, and the presence of HClO₄ appears to be critical for initiating the energy-

autonomous carbonization. Therefore, does the doping level of HClO₄ in PANI affect the reaction temperature and further affect the formation of the resulting amorphous carbon? How can the HClO₄ doping level be controlled? (2) The synthesized PANI fibers were centrifuged from the aqueous phase and dried in air at 60 °C for 22 hours. However, they still exhibited a water content of 60–160% when transferred to the three-neck flask for reaction. Could the authors clarify why such high water retention occurred, and how the water content can be effectively controlled? The manuscript also mentions that the successful execution of the popping reaction is also highly dependent on the presence of a specific quantity of water within the PANI matrix. A quantitative analysis is needed to evaluate whether different levels of water content influence the occurrence of the reaction and the formation of amorphous carbon.

Response: We are deeply grateful to the reviewer for these insightful questions, which have been instrumental in guiding us toward a more precise understanding of the reaction mechanism. Our initial hypothesis indeed identified water and doped HClO₄ as key factors. However, prompted by the reviewer's call for a more quantitative analysis, we conducted a systematic investigation of the precursor composition and reaction conditions, including precursor amount, heating rate, water content, and the content/type of HClO₄. This comprehensive study revealed that free HClO₄, rather than water or doped HClO₄, is the critical component for triggering the popping reaction. In other words, the reactive precursor is most accurately described as a polyaniline-HClO₄ composite. Water is typically present because it forms hydrates with free HClO₄, but it is not an independent prerequisite for the reaction to occur. This new understanding directly addresses the reviewer's specific questions:

HClO₄ doping level: The reaction initiation temperature and product morphology are primarily governed by the free HClO₄ content, which are the residue HClO₄ after water wash. The doping level of HClO₄ in the PANI backbone itself is less critical for the popping reaction. The free HClO₄ content can be precisely controlled through adjusting the HClO₄ concentration during PANI synthesis, modifying the post-synthesis washing protocols, or via the deliberate addition of exogenous HClO₄ solution to the synthesized polymer.

Water content: The previously reported high water retention of 60-160% resulted from an analytical oversight, as the initial quantitative model did not fully account for the HClO₄ content. Using our revised and more accurate protocol (detailed in Supplementary Note 4), we determined the typical water content in a reactive precursor to be approximately 20%. This water content can be controlled via drying time, and its primary function is to modulate reaction intensity by influencing thermal transport, rather than serving as an essential prerequisite for the popping reaction.

We have thoroughly revised the Abstract and Results and Discussion sections in the manuscript (Page 2, 13, 15) to reflect this refined mechanism and have provided a detailed quantitative analysis of all reaction factors in Supplementary Note 4 (Pages 28-60 in the Supplementary Information).

Page 2 in the revised manuscript:

Here, we introduce an energy-autonomous pathway that utilizes the intrinsic chemical energy stored in a polyaniline-HClO₄ composite, bypassing conventional energy barriers. Upon mild thermal, microwave, or mechanical stimulation, the precursor undergoes an ultrafast (~0.4 s) exothermic self-propagation driven by the explosive decomposition of free HClO₄. This single step simultaneously generates intense localized heat and an explosive volume of gas, which forcibly exfoliates and carbonizes the polyaniline into interconnected two-dimensional amorphous carbon nanosheets. We rigorously validate that this energy-efficient method achieves a carbon conversion efficiency comparable to conventional pyrolysis and demonstrate that the reaction vigor is precisely tunable via precursor water content, confirming its strong potential for safe scale-up.

Page 14-15 in the revised manuscript:

Based on these findings, we propose a synergistic mechanism for the energy-autonomous carbonization and exfoliation of PANI. As illustrated in Supplementary Fig. 53, the transformation requires a PANI emeraldine salt containing a critical amount of free HClO₄. Upon heating, HClO₄ decomposes, initiating a sequence of events. First, the decomposition

generates potent radical species and oxidants, such as ClO_2 , ClO , and O_2 ⁴¹, triggering a violent, highly exothermic oxidation of PANI, which has long been recognized as an effective radical scavenger/heat insulator, and can be used to desensitize nanothermites⁴²⁻⁴⁴. The decomposition of perchlorate species also concomitantly releases a large volume of gases (e.g., Cl_2 , O_2 , CO_2 , H_2O). The instantaneous release of energy drives rapid dehydrogenation and carbonization, with local temperature confirmed to exceed 1400 °C (Supplementary Note 4). Simultaneously, the violent gas evolution creates a critical internal pressure that physically exfoliates the carbonizing polymer matrix from the inside out, while the in-situ gas-bubble templating directly yields the characteristic 2D porous nanosheet morphology with an open, interconnected pore network (Supplementary Fig. 47). This synergy between intense localized heating and explosive gas expansion culminates in a thermal runaway event, observed as a rapid “pop”. While the reaction initiation temperature and overall POP-C yield are governed by external factors (e.g., PANI packing density and amount, heating rate, water content) as detailed in Supplementary Note 4, the successful formation of the desired 2D nanosheet morphology critically requires a high free HClO_4 content (mass content >30%), which ensures sufficient gas volume and exothermicity for complete carbonization and exfoliation.

Page 15 in the revised manuscript:

In stark contrast, the popping reaction described herein is characterized by the decomposition of energetic components in the PANI precursor, triggering rapid carbonization in less than one second. Concurrently, vigorous outgassing promotes the formation of 2D carbon nanosheet. The popping process exhibits strong potential for scale-up due to its speed, low external energy demand, and tunable reaction intensity, which can be moderated via precursor water content, as detailed in Supplementary Notes 4. Moreover, the yield of POP-C from the popping reaction ranges from 17% to 40% depending on reaction conditions, which is comparable to that of conventional pyrolysis (37%), as detailed in Supplementary Notes 4 and 5.

Page 17 in the revised manuscript:

In summary, we present a rapid, energy-autonomous synthesis of 2D amorphous carbon nanosheets with high surface area and atomically dispersed metal sites. This method leverages

the intrinsic chemical energy stored in a polyaniline-HClO₄ composite. Upon mild heating, the composite undergoes a popping reaction that simultaneously generates a large amount of heat and gas, carbonizing and exfoliating the polymer into porous nanosheets almost instantaneously. Furthermore, the precursor design enables precise incorporation of transition metals (e.g., Fe, Co, Ni) during synthesis, allowing the creation of tailored active sites that enhance performance in electrocatalytic reactions. We believe this versatile, energy-efficient approach establishes a new pathway for the rapid fabrication of functional 2D carbons, with promising applications extending from electrocatalysis to energy storage and beyond.

Supplementary Note 4 in the Supplementary Information (Pages 28-60):

Supplementary Note 4: Key factors and mechanism governing the popping reaction

To elucidate the mechanism of the popping reaction that converts PANI to POP-C, we first analyzed the composition of viable PANI precursors. We then systematically investigated the effects of several key parameters on the reaction: heating rate, precursor mass, water content, HClO₄ concentration during PANI synthesis, and the type of HClO₄ within the precursor. The collective influence of these factors provides critical insight into the underlying reaction mechanism.

1. The composition of PANI precursor

Our investigation reveals that the popping reaction, which simultaneously exfoliates and carbonizes PANI into POP-C at low initiating temperatures (~120 °C), has strict precursor requirements. This transformation occurs exclusively in PANI synthesized in HClO₄ solution and subsequently washed with water (Supplementary Fig. 1c). PANI emeraldine salt can be prepared in various acid aqueous solution, such as hydrochloric acid solution (PANI-HCl) or sulfuric acid solution (PANI-H₂SO₄), yet these variants do not undergo the popping reaction (Supplementary Fig. 31), underscoring the unique role of the perchlorate acid.

Supplementary Fig. 31 Schematic showing the possible factors that influence the popping reaction of PANI fibers under moderate initiating temperature.

Furthermore, the post-synthesis treatment is critical. Water-washed PANI retains a wet, sticky state (Supplementary Fig. 1c), indicating the presence of residual water and, presumably, HClO_4 , originated from the residual aqueous HClO_4 solution in the PANI. Even after long time drying process at $60\text{ }^\circ\text{C}$, the PANI cannot be totally dried. In contrast, PANI washed with acetone forms a dry, green powder (PANI-D), suggesting the effective removal of these components (Supplementary Fig. 1c). Water-washed PANI are able to be popped and carbonization to form POP-C at low initiating temperature (Supplementary Fig. 2 and 3). The resulting POP-C exhibits exceptional thermal stability. Even after annealing at $800\text{ }^\circ\text{C}$ for 2 h under N_2 (denoted as POP-C800), the material maintains its original fluffy morphology and ultralow density (Supplementary Fig. 32a-d). While PANI-D maintains a similar fibrous morphology to standard PANI (Supplementary Fig. 32e), it does not undergo the popping reaction. Heating PANI-D to $150\text{ }^\circ\text{C}$ results in a dense material (PANI-D150) that retains its short fibrous structure (Supplementary Fig. 32f). Subsequent high-temperature pyrolysis ($800\text{ }^\circ\text{C}$) of PANI-D yields a dense, agglomerated black powder (PANI-D800, Supplementary Fig. 31g-h), starkly contrasting with the fluffy, nanosheet morphology of POP-C obtained from the popping reaction (Supplementary Fig. 32b-d). These findings highlight the crucial factors

for the low-temperature popping reaction and raise central mechanistic questions: Why is HClO₄-doped PANI uniquely capable of this transformation? What specific components are removed by the acetone washing process that deactivate the popping reaction?

Supplementary Fig. 32 SEM images of (a) PANI, (b) POP-C and (c) POP-C800. (d) Volume comparison of 200 mg PANI, 20 mg POP-C and 20 mg POP-C800. SEM images of (f) PANI-D, (g) PANI-D150 and (h) PANI-D800. (e) Volume comparison of 200 mg PANI-D, 160 mg PANI-D150 and 50 mg PANI-D800.

As detailed in the Experimental section and Supplementary Fig. 1, the PANI precursor for POP-C was synthesized via interfacial polymerization of aniline. The oxidative polymerization

occurs at the interface, forming polyaniline, which subsequently migrates into the aqueous phase (1.0 M HClO₄). The resulting product, confirmed to be emeraldine salt doped with HClO₄, was separated from the aqueous phase by centrifugation, washed with water, and air-dried at 60 °C for 22 h. The obtained PANI was in a wet, sticky state, indicating the likely presence of multiple components: the HClO₄ doped PANI emeraldine salt itself, residual free HClO₄ from the excess acid in the aqueous phase, and water. To quantitatively analyze these compositions, we developed a straightforward method based on acid-base titration, as illustrated in Supplementary Fig. 33.

Supplementary Fig. 33 Quantification protocol for the composition of the PANI precursor, including polyaniline base, doped HClO₄, free HClO₄, and water contents.

First, a known mass of the PANI sample was dispersed in an excess of 0.1 M KOH to ensure complete neutralization of all HClO₄ species. This includes both the doped HClO₄ (which forms a salt with the polyaniline backbone) and the free HClO₄ (residual acid from synthesis). The chemical reaction, in which the emeraldine salt is converted to the un-doped emeraldine base, is illustrated in Supplementary Fig. 34. Following this reaction, the PANI suspension turns from green to dark blue, visually confirming the formation of the emeraldine base (Supplementary Fig. 35a). After approximately 6 h, the PANI emeraldine base precipitates. The supernatant, containing the unreacted KOH, was then transferred to a beaker and titrated with 0.1 M HClO₄ using phenolphthalein (PhPh) as an indicator (Supplementary Fig. 35a). The

amount of HClO_4 consumed in the initial neutralization was determined by calculating the difference between the initial KOH and the amount remaining after reaction, as quantified by the titration.

Supplementary Fig. 34 Schematic of the quantitative analysis for free and doped HClO_4 in PANI. (a) The PANI emeraldine salt is reacted with KOH . Titration of consumed KOH quantifies total HClO_4 (free and doped HClO_4). (b) The PANI emeraldine salt is dispersed in water. Titration of the aqueous phase quantifies free HClO_4 . Doped HClO_4 is calculated by difference.

Supplementary Fig. 35 Quantitative analysis of HClO₄ in PANI. (a) Measurement of total HClO₄ content. (b) Measurement of free HClO₄ content. The doped HClO₄ content is the difference of total and free HClO₄ content.

Second, a known mass of the PANI sample was dispersed in water to dissolve the free HClO₄ (Supplementary Fig. 34) The suspension remained green, characteristic of the PANI emeraldine salt (Supplementary Fig. 35b). After approximately 6 h, the PANI emeraldine salt precipitated. The supernatant, which contained the dissolved free HClO₄, was transferred to a beaker and titrated with 0.1 M KOH using PhPh as an indicator (Supplementary Fig. 35b). The amount of free HClO₄ was directly quantified from the titration results. The mass of doped HClO₄ was subsequently calculated by subtracting the free HClO₄ content from the total HClO₄ content.

Third, a control sample of polyaniline without water and free HClO₄, designated PANI-D, was prepared. PANI-D was synthesized via the same interfacial polymerization of aniline as the standard PANI. The critical difference lies in the washing procedure: after separation from the aqueous phase, the polymer was washed with acetone to remove all water and free HClO₄, followed by air-drying at 60 °C for 22 h. The free HClO₄ originates from the residual HClO₄ solution trapped within the standard PANI matrix. Acetone washing effectively removes both this residual acid and the associated water. Consequently, PANI-D is obtained as a green powder, in contrast to the standard PANI, which contains water and free HClO₄ and presents as a black, sticky solid (Supplementary Fig. 1). The mass of polyaniline without doped HClO₄ (PANI base) was subsequently calculated by subtracting the doped HClO₄ content from the PANI-D content.

Finally, the water content can be obtained by subtracting the doped HClO₄ content, free HClO₄ content and PANI base content from the PANI content.

To investigate how the heating rate, precursor mass, water content, HClO₄ content, and the type of HClO₄ within the precursor influence the popping reaction, several PANI precursor were prepared and the PANI emeraldine base, doped HClO₄, free HClO₄ and water content was quantified based on the protocol in Supplementary Fig. 33. The quantitative compositional analysis via titration was performed at least three times for each sample to ensure reproducibility and to generate error bars.

2. Influence of the PANI precursor amount

The PANI used to investigate the influence of precursor amount was synthesized using 1.0 M HClO₄ as the aqueous phase, followed by water washing and drying at 60 °C for 22 h. Its composition was quantified as 32% PANI emeraldine base, 15% doped HClO₄, 33% free HClO₄, and 20% water. Video recordings of the popping process for different PANI masses (50, 100, and 200 mg) at 15 °C min⁻¹ are provided in Supplementary Video 5. Key frames from these videos, highlighting critical stages of the reaction, are compiled in Supplementary Fig. 36. Using 200 mg of PANI precursor led to a fierce reaction, as indicated by a bright flame

observed within the first 33 ms (Supplementary Fig. 36a). The flame subsequently subsided and vanished completely by 500 ms. When the PANI mass was reduced to 100 mg, the intensity of the popping reaction decreased (Supplementary Fig. 36b). A further reduction to 50 mg resulted in an even less intense reaction (Supplementary Fig. 36c). These observations confirm that while the popping reaction to produce POP-C proceeds across different PANI masses, the reaction intensity diminishes significantly with decreasing precursor amount.

Supplementary Fig. 36 Synthesis of POP-C via rapid thermal popping at a heating rate of $15\text{ }^{\circ}\text{C min}^{-1}$ with varying amounts of PANI precursor: (a) 200 mg, (b) 100 mg, and (c) 50 mg. The temperature displays alternates between the current thermocouple reading (actual temperature) and the set-point. The popping reaction initiates at the thermocouple reading (e.g., $127\text{ }^{\circ}\text{C}$, as shown in panel b), not at the displayed set-point value.

Our investigation into the influence of PANI mass revealed that the physical packing of the precursor is a critical factor for the popping reaction. Specifically, we compared two configurations for 50 mg of PANI in a round-bottom flask: a densely packed state and an intentionally scattered one. To create the packed state, the PANI was concentrated at the base of the flask using tapping and vibration (Supplementary Fig. 37a). This configuration underwent a successful popping reaction at an initiation temperature of 143 °C, yielding fluffy POP-C. In contrast, when the same mass of PANI was intentionally scattered across the flask bottom (Supplementary Fig. 37b), no popping reaction occurred, even when heated to 243 °C. These results demonstrate that a densely packed precursor configuration is essential for the popping reaction, as it facilitates the heat accumulation required to trigger thermal runaway.

Supplementary Fig. 37 Effect of PANI configuration on the popping reaction. (a) Heating 50 mg of packed PANI induces a popping reaction upon reaching 143 °C. (b) In contrast, heating 50 mg of scattered PANI to 243 °C results in no popping reaction.

The reaction initiation temperature and final POP-C yield are both dependent on the mass of the PANI precursor. As shown in Supplementary Fig. 38a, the initiation temperature decreases

from 124 °C to 129 °C to 146 °C as the PANI mass is reduced from 200 mg to 100 mg to 50 mg. This trend is likely caused by the more densely packed configuration achieved with a larger precursor mass, which facilitates more efficient heat accumulation and thus lowers the energy required to initiate the reaction. The yield of POP-C was calculated based on the mass of the PANI emeraldine base in the precursor. This is because only the emeraldine base component, excluding the doped HClO₄, free HClO₄ and water, is considered as the carbon precursor converted into the final carbonaceous product. For the PANI used in this series of experiments (comprising 32% PANI base, 15% doped HClO₄, 33% free HClO₄, and 20% water), the yield decreased from 40% to 32% to 28% with increasing precursor mass (Supplementary Fig. 38b). This trend is attributed to product loss during the violent reaction; a portion of the POP-C is expelled from the flask by the rapid gas generation and cannot be collected. Since the reaction intensity diminishes with lower precursor mass (as shown in Supplementary Fig. 36), the relative product loss is reduced, leading to a higher relative yield at lower masses.

Supplementary Fig. 38 Effect of PANI precursor mass on the popping reaction and resulting POP-C. (a) The reaction initiation temperature as a function of PANI mass. (b) The final yield of POP-C vs the initial PANI mass. (c) CV curves and (d) Raman spectra of POP-C synthesized from different PANI masses.

The electrochemical surface area (ECSA) of the POP-C samples, synthesized with different PANI precursors, was evaluated by cyclic voltammetry (CV) in 0.1 M KOH. The measurements were performed within a potential range of 0.65 to 1.05 V vs RHE, which corresponds to the capacitive, non-Faradaic double-layer region (Supplementary Fig. 38c). In this region, the current is predominantly capacitive, and its magnitude is directly proportional to the ECSA. The highly rectangular shape of the CV curves, maintained even at a high scan rate of 100 mV s⁻¹, indicates a well-interconnected, open pore structure that facilitates rapid ion transport. The near-identical CV profiles observed for all POP-C samples, regardless of the

PANI precursor amount, suggest that they possess comparable surface areas and porous architectures. Raman spectra of POP-C synthesized from different precursor masses are comparable (Supplementary Fig. 38d), indicating that the precursor loading does not influence the carbon hybridization, defect density, or level of disorder in the final product.

3. Influence of the heating rate

The popping reaction was conducted using an IKA RCT basic heating mantle with a metal bath. The heating rate was controlled by setting a target temperature, with higher target temperatures resulting in faster heating rates (Supplementary Fig. 39). For instance, setting the target temperature to 300 °C and 200 °C yielded heating rates of 22 °C min⁻¹ (in the range of 85 to 150 °C) and 15 °C min⁻¹ (in the range of 100 to 150 °C), respectively. To achieve a slower heating rate, a stepped protocol was employed: the target temperature was initially set to 100 °C, then increased to 150 °C upon reaching 80 °C, to 180 °C upon reaching 130 °C, and finally to 210 °C upon reaching 160 °C. This stepped protocol lead to a heating rate of stepped 6 °C min⁻¹ in the range of 110 to 160 °C.

Supplementary Fig. 39 Heating rates for the IKA RCT basic heating mantle. The heating rate was calculated from a linear fit of the temperature vs time curve. The fitting range was selected based on the typical reaction temperature range: 85-150 °C for a rate of 6 °C min⁻¹, 100-150 °C for 15 °C min⁻¹, and 110-160 °C for 22 °C min⁻¹.

The PANI used to investigate the influence of heating rate was synthesized using 1.0 M HClO₄ as the aqueous phase, followed by water washing and drying at 60 °C for 22 h. Its composition was quantified as 32% PANI base, 15% doped HClO₄, 33% free HClO₄, and 20% water. Video analysis of POP-C synthesis from 100 mg PANI (Supplementary Video 6, key frames in Supplementary Fig. 40) reveals that the reaction becomes more intense at higher heating rates. The flame observed at 22 °C min⁻¹ is significantly brighter than those at 15 and 6 °C min⁻¹, indicating a more violent reaction.

Supplementary Fig. 40 Synthesis of POP-C from PANI via rapid thermal popping at different heating rates: (a) 6 °C min⁻¹, (b) 15 °C min⁻¹, and (c) 22 °C min⁻¹. The temperature displays alternates between the thermocouple reading (the actual temperature) and the set-point temperature. The popping reaction initiates at the thermocouple reading (e.g., 127 °C in panel b), not at the set-point value.

Both the reaction initiation temperature and the final POP-C yield are influenced by the heating rate. As shown in Supplementary Fig. 41a, the initiation temperature decreases from 139 °C to 129 °C and finally to 113 °C as the heating rate is increased from 6 to 15 to 22 °C min⁻¹. This trend occurs because a higher heating rate facilitates more efficient heat accumulation, thereby lowering the initiation temperature of the reaction. The product yield decreases from 34% to 32% to 26% with increasing heating rates (Supplementary Fig. 41b). This inverse relationship between yield and heating rate is attributed to greater product loss during the more violent reactions; a larger portion of the POP-C is expelled from the flask by the rapid gas generation at higher heating rates and cannot be collected. This explanation is consistent with the observed increase in reaction intensity (i.e., brighter flame) at higher heating rates, as shown in Supplementary Fig. 40. The lowest triggering temperature observed was 86 °C, using 200 mg of PANI precursor and a heating rate of ~30 °C min⁻¹. ***However, combining a large precursor mass with a high heating rate can produce an excessively violent reaction. Therefore, this specific combination is not recommended due to safety concerns.***

Supplementary Fig. 41 Effect of heating rate on the popping reaction and resulting POP-C.

(a) The dependence of reaction initiation temperature on PANI mass. (b) POP-C yield as a function of PANI mass. (c) CV curves and (d) Raman spectra of POP-C synthesized at different heating rates.

The ECSA of the POP-C samples, synthesized with different heating rates, was evaluated by CV in 0.1 M KOH. The highly rectangular shape of the CV curves, maintained even at a high scan rate of 100 mV s^{-1} , indicates a well-interconnected, open pore structure that facilitates rapid ion transport (Supplementary Fig. 41c). The near-identical CV profiles across all samples suggest that the heating rate during synthesis has a negligible influence on the resulting surface area and porous architecture. The comparable Raman spectra of POP-C synthesized under different heating rates (Supplementary Fig. 41d) indicate that this parameter has no significant influence on the carbon hybridization, defect density, or structural disorder of the final product.

4. Influence of the water content in PANI

The PANI used to investigate the influence of water content was synthesized using 1.0 M HClO₄ as the aqueous phase, followed by water washing. To achieve different water contents, the drying time at 60 °C was systematically varied from 46 to 22 to 8 to 4 h, producing PANI with final water contents of 11%, 20%, 35%, and 58%, respectively. Precursors with these different water contents were then loaded into the reaction flask for the popping process. To ensure a constant mass of solid material (PANI base, doped HClO₄ and free HClO₄) across all experiments, the total PANI mass was adjusted to 180 mg (11% water), 200 mg (20% water), 250 mg (35% water), and 400 mg (58% water). As shown in Supplementary Video 7 and Fig. 42, the intensity of the popping reaction decreases as the water content in the PANI precursor increases. The popping process for PANI with 11% and 20% water content produces a significantly brighter flame than that observed for samples with 35% and 58% water, indicating a more violent reaction at lower hydration levels.

Supplementary Fig. 42 Synthesis of POP-C from PANI via rapid thermal popping with varying water contents: (a) 11%, (b) 20%, (c) 35%, and (d) 58%. The temperature displays alternates between the set-point and the actual thermocouple reading. The popping reaction initiates at the thermocouple reading (e.g., 163 °C in panel c, 184 °C in panel d), not at the displayed set-point value (e.g., 250 °C).

Both the reaction initiation temperature and the final POP-C yield are influenced by the water content. As shown in Supplementary Fig. 43a, the initiation temperature increases from 114 °C to 124 °C to 164 °C and finally to 186 °C as the water content increases from 11% to 20% to 35% to 58%. During the heating process, water gradually evaporates from all types of PANI precursors before the popping reaction occurs. This evaporation is an endothermic process that cools the system. For PANI with higher water content, this heat loss is more severe, thereby hindering efficient heat accumulation. As discussed in the sections on the “Influence of PANI amount” and “Influence of the heating rate”, such heat accumulation is a critical factor for triggering the popping process. Consequently, precursors with higher water content require more time and a higher ultimate temperature to fully dehydrate and accumulate the substantial heat necessary to initiate the ignition reaction. Furthermore, high water content adversely affects the packing density, as the sticky, hydrated PANI adheres to the flask walls rather than consolidating into a dense mass. The product yield increases from 17% to 28% to 33% and finally to 36% with increasing water content (Supplementary Fig. 43b). This positive correlation is attributed to the less violent popping process observed at higher hydration levels. As the water content increases, the reaction intensity diminishes, resulting in less POP-C being expelled from the flask by rapid gas generation. The milder reaction thus minimizes product loss and leads to a higher collected yield.

A key consideration for scaling the popping reaction is managing its vigor. We have demonstrated that the reaction intensity can be effectively modulated by adjusting the water content of the precursor; higher water content reduces the intensity, albeit at the cost of a higher triggering temperature. *Therefore, precursors with elevated water content present a viable pathway for safer scale-up.*

Supplementary Fig. 43 Effect of water content in PANI precursor on the popping reaction and resulting POP-C. (a) The reaction initiation temperature and (b) final yield of POP-C as a function of water content. (c) CV curves and (d) Raman spectra of POP-C synthesized from PANI precursor containing different water

The ECSA of the POP-C samples, synthesized with PANI contained different water amount, were evaluated by CV in 0.1 M KOH. The highly rectangular shape of the CV curves, maintained even at a high scan rate of 100 mV s⁻¹, indicates a well-interconnected, open pore structure that facilitates rapid ion transport (Supplementary Fig. 43c), suggesting that the precursor's hydration level does not affects the resulting POP-C's porous architecture. The comparable Raman spectra of POP-C synthesized from PANI precursors with different water contents (Supplementary Fig. 43d) indicate that this parameter has no significant influence on the carbon hybridization, defect density, or structural disorder of the final product.

5. Influence of the HClO₄ concentration during PANI synthesis

To investigate the influence of HClO₄ content on the POP-C synthesis, PANI samples were prepared using HClO₄ at various concentrations (0.1, 0.3, 0.5, 0.7, 1.0, 1.5 M), followed by water washing and drying. The PANI's composition was quantified using the titration protocol detailed in Supplementary Fig. 33. As illustrated in Fig. 5b, the HClO₄ content in PANI varies significantly with the concentration of the synthesis solution. Specifically, both the total HClO₄ content (doped and free) and the free HClO₄ content increase with the HClO₄ concentration. In contrast, the doped HClO₄ content exhibits a volcano-shaped trend, initially increasing and then decreasing slightly beyond 0.7 M. It is important to note that this slight decrease in the doped HClO₄ content from 0.7 M to 1.5 M does not reflect a lower doping degree of the polymer backbone. Instead, it is attributed to a decrease in the relative mass fraction of the PANI base itself within the precursor, caused by the increasing proportion of free HClO₄ (Supplementary Table 4).

Supplementary Table 4 Composition of PANI synthesized using different HClO₄ concentrations.

[HClO ₄] (M)	Total HClO ₄ (%)	Free HClO ₄ (%)	Doped HClO ₄ (%)	PANI base (%)	water (%)
0.10	7	4	3	66	20
0.30	20	12	8	61	19
0.50	30	17	14	46	23
0.70	38	21	17	40	22
1.00	49	33	15	32	20
1.50	60	47	13	22	18

Since the PANI-D (containing only doped HClO₄ but no free HClO₄) is not able to undergo popping reaction. We propose that the free HClO₄ is the reaction key factor that trigger the popping reaction to form POP-C. To investigate the influence of HClO₄ content on the popping reaction, 200 mg PANI with varying free HClO₄ content (4%, 12%, 17%, 21%, 33%, 47%) were used to carry the popping reaction. As shown in Supplementary Fig. 44, PANI with a very low HClO₄ content (4%) does not undergo the popping reaction, even when heated to 247 °C.

When the HClO_4 content is increased to 12%, a small-scale popping reaction occurs, but it lasts for only ~ 200 ms (Supplementary Fig. 45a and Video 8). As the HClO_4 content is raised further to 17% and 21%, the reaction intensifies, producing a brighter and larger flame that persists for approximately 500 ms (Supplementary Fig. 45b-c and Video 8). At the highest concentrations of 33% and 47%, the reaction becomes violently explosive, with flames rushing out of the flask (Supplementary Fig. 45d-e and Video 8). The reaction duration also increases, lasting ~ 700 ms for the 47% free HClO_4 sample (Supplementary Fig. 45e and Video 8). These observations collectively demonstrate that the intensity and violence of the popping reaction are directly correlated with the HClO_4 content in the PANI precursor.

Supplementary Fig. 44 Heating 200 mg of PANI with total HClO_4 contents of 7% (free HClO_4 contents of 4%) to 247 °C results in no popping reaction.

Supplementary Fig. 45 Synthesis of POP-C from 200 mg of PANI via rapid thermal popping with varying free HClO₄ contents: (a) 12%, (b) 17%, (c) 21%, (d) 33%, and (e) 47%.

The reaction initiation temperature decreases significantly with increasing the free HClO₄ content (Supplementary Fig. 46a), suggesting that free HClO₄ is the key driver of the popping reaction. We propose that the mechanism involves the thermal decomposition of concentrated HClO₄, which produces chlorine, oxygen, water, and releases a significant amount of heat. The rapid release of these gases constitutes an explosive event. At lower HClO₄ concentrations, the acid is highly dispersed within the PANI matrix, leading to a moderated decomposition that requires a higher initiation temperature. In contrast, a higher HClO₄ content creates localized concentrations of acid within the precursor, facilitating a more violent decomposition that initiates at a lower temperature. This effect is so pronounced that unwashed PANI, with its extremely high HClO₄ content, can undergo a popping reaction at room temperature during grinding (Supplementary Fig. 3). Correspondingly, the final POP-C yield decreases with increasing the free HClO₄ content (Supplementary Fig. 46b). This inverse relationship is attributed to greater product loss during the more violent reactions; a larger portion of POP-C is expelled from the flask by rapid gas generation and cannot be collected. As shown by the flame during the popping process in Supplementary Fig. 45, PANI containing 12% and 17% free HClO₄ content presents reddish flame, which PANI containing 33% and 47% free HClO₄ content present yellow flame, while the PANI containing 21% present orange color, indicating that the reaction temperature of the popping moment increase with the HClO₄ content.

Supplementary Fig. 46 Effect of HClO₄ content in PANI precursor on the popping reaction and resulting POP-C. (a) The reaction initiation temperature as a function of the free HClO₄ content. (b) The final yield of POP-C as a function of free HClO₄ content. (c) CV curves and (f) Raman spectra of POP-C synthesized from PANI precursors with varying free HClO₄ content.

Safety Warnings:

Polyaniline (PANI) composites containing elevated levels of HClO₄, especially free acid, are highly reactive and may combust spontaneously. To mitigate this risk, the preparation of precursors with total HClO₄ content >60% (free HClO₄ >47%) is strongly discouraged!

The ECSA of POP-C samples synthesized from PANI with different HClO₄ contents was

evaluated by CV in 0.1 M KOH. POP-C derived from PANI with high free HClO₄ content (33% and 47%) exhibits highly rectangular CV curves, even at a high scan rate of 100 mV s⁻¹, indicating a well-interconnected, open pore structure that facilitates rapid ion transport (Supplementary Fig. 46c). The comparable curves for these two samples suggest they possess similar ECSA values. POP-C from a medium free HClO₄ content (21%) also shows a rectangular shape but with a reduced double-layer current, indicating good conductivity yet a smaller ECSA. In contrast, POP-C from PANI with lower HClO₄ content (12% and 17%) shows a significantly smaller double-layer current and distorted, “leaf-like” CV profiles. This shape suggests high electrical resistance, mass transport limitations, a much smaller surface area, and a less developed pore structure. These inferior electrochemical properties are a direct result of incomplete exfoliation, caused by the smaller amount of gas generation associated with insufficient HClO₄ content.

Raman spectroscopy of the popping products (Supplementary Fig. 46d) confirms the formation of carbonaceous materials rather than PANI. With the increase of HClO₄ content from 12% to 17% to 21%, the I_D/I_G value increases from 0.85 to 0.87 to 0.95, while when the HClO₄ content increases from 21% to 33% to 47%, the I_D/I_G stays at the same level (0.95, 1.00 and 0.98). We propose that this trend is governed by the extent of exfoliation during the popping reaction. This reaction induces rapid dehydration of the polyaniline precursor, promoting the graphitization of its benzene rings into sp² carbon. Higher HClO₄ content elevates the reaction temperature and gas evolution, facilitating the formation and growth of sheet-like carbon. For precursors with low free HClO₄ content (12% and 17%), carbonization occurs without full exfoliation, resulting in a less disordered structure and a lower I_D/I_G ratio. In contrast, for precursors with high HClO₄ content (33% and 47%), carbonization is accompanied by complete exfoliation, yielding sheet-like nano-carbons with an abundance of crystallite boundaries, which is consistent with the observed higher I_D/I_G ratios³⁵.

Further TEM characterization of the POP-C samples revealed a clear morphological evolution directly correlated with the HClO₄ content (Fig. 5d-h). The POP-C derived from a precursor with 12% free HClO₄ content exhibits an agglomerated morphology (Supplementary Fig. 5d),

indicating that the heat generated was sufficient for carbonization but the generated gas is insufficient for exfoliation. As the HClO_4 content increased to 17%, the resulting POP-C began to show the presence of nanosheets alongside agglomerates (Fig. 5e), signifying the onset of partial exfoliation. At 21% HClO_4 , the product displayed an interconnected, thin-sheet structure, pointing to a more advanced exfoliation process (Fig. 5f). Finally, precursors with high HClO_4 content (33% and 47%) yielded POP-C consisting entirely of well-defined, 2D interconnected nanosheets (Fig. 5g-h). This systematic progression confirms that the violent gas evolution during the popping reaction is the critical factor for exfoliation. These observations provide direct visual evidence for the proposed formation mechanism: the popping reaction simultaneously drives rapid carbonization through intense local heating and forcibly exfoliates the intermediate through internal gas pressure, dynamically separating the polymer matrix into large-area, porous 2D nanosheets. This process also creates an overall porous foam morphology (Supplementary Fig. 47), resulting from the synergy between explosive exfoliation and in-situ gas-bubble templating. Consequently, the generation of gas (directly governed by the HClO_4 content in the precursor) is the fundamental driver of the unique 2D nanosheet architecture.

Supplementary Fig. 47 TEM image of Ni-POP-C, showing evidence of gas-bubble templating.

6. Further verify the function of free HClO₄

As established previously, the PANI precursor, comprising the emeraldine base, doped HClO₄, free HClO₄, and water, can undergo a popping reaction to yield POP-C. HClO₄ exists in two distinct states within the precursor: doped HClO₄, which forms the PANI emeraldine salt (Fig. 5a and Supplementary Fig. 33), and free HClO₄, which remains due to incomplete washing. To deconvolute their individual roles and that of water, we synthesized PANI-D using 1.0 M HClO₄ as the aqueous phase, followed by acetone washing and drying at 60 °C for 22 h. This acetone wash removes all residual aqueous solution (both water and free HClO₄), yielding a precursor composed solely of the HClO₄ doped PANI salt. As shown in Supplementary Fig. 48a, this PANI-D sample, containing no free HClO₄ and no water, did not undergo the popping reaction. To isolate the role of water, it was intentionally reintroduced by dispersing PANI-D in water and drying it for 1.5 h under 60 °C (Supplementary Fig. 49a). The water content of this hydrated sample was determined to be 20% by measuring the mass difference before and after water addition. However, this hydrated PANI-D containing both water and doped HClO₄ also failed to undergo the popping process (Supplementary Fig. 48b), demonstrating that water alone is not sufficient to drive the popping reaction and that free HClO₄ is an essential factor.

Supplementary Fig. 48 The necessity of free HClO₄ for the popping reaction. (a) PANI containing no free HClO₄ and no water shows no reaction upon heating to 236 °C. (b) Similarly,

PANI with no free HClO_4 but with 20% water content also shows no reaction upon heating to $227\text{ }^\circ\text{C}$.

Supplementary Fig. 49 (a) Introduction of water to dry HClO_4 -doped PANI (PANI-D). (b) Introduction of free HClO_4 to PANI-D. (c) Introduction of free HClO_4 to dry HCl-doped PANI (PANI-HCl-D).

To test the hypothesis that free HClO_4 , rather than doped HClO_4 or water, is the key factor for the popping reaction, we intentionally introduced it into the PANI-D matrix. Specifically, 120 mg of PANI-D was dispersed in 0.6 mL of 1 M HClO_4 aqueous solution and dried at $60\text{ }^\circ\text{C}$ for 3 h, yielding 200 mg of PANI precursor containing both water and free HClO_4 (Supplementary Fig. 49b). The mass of the introduced free HClO_4 was calculated to be 60.3 mg ($0.6\text{ mL} \times 1\text{ mol L}^{-1} \times 100.46\text{ g mol}^{-1}$), resulting in a free HClO_4 content of 30% relative to the total precursor mass (200 mg). This value is comparable to the intrinsic free HClO_4 content (33%) found in standard PANI synthesized in 1 M HClO_4 and washed with water. To confirm that free

HClO₄ can activate a different PANI salt, we synthesized PANI-HCl-D via interfacial polymerization using 1 M HCl, followed by acetone washing. This resulted in a dry powder containing only PANI base and doped HCl, without water or free acid. Free HClO₄ was then introduced to this matrix by dispersing 90 mg of PANI-HCl-D in 0.6 mL of 1 M HClO₄ solution, followed by drying at 60 °C for 3 h (Supplementary Fig. 49c). As shown in Supplementary Video 9 and Fig. 50, both PANI-D and PANI-HCl-D with added free HClO₄ underwent successful popping reactions at approximately 120 °C, yielding POP-C. The intensity of these reactions was comparable to that of standard PANI containing 33% intrinsic free HClO₄.

Supplementary Fig. 50 Synthesis of POP-C from 200 mg PANI via rapid thermal popping under different acid conditions. (a) PANI with 33% intrinsic free HClO₄. (b) HClO₄-doped

PANI with additional free HClO₄ introduced. The content of the free HClO₄ is 30%. (c) HCl-doped PANI with free HClO₄ introduced. The content of the free HClO₄ is 33%. The temperature displays alternates between the set-point and the actual thermocouple reading. The popping reaction initiates at the thermocouple reading (e.g., 119 °C in panel b), not at the displayed set-point value (e.g., 200 °C).

During the reaction, chlorine can be detected in the fume hood by using a chlorine gas sensor, which is due to the decomposition of HClO₄. The decomposition of HClO₄ can also generate H₂O and O₂:

In the presence of polyaniline, additional gases are formed due to oxidation at high local temperatures. IR spectroscopy confirms the release of CO₂ (Supplementary Fig. 51), resulting from the strong oxidative carbonization of the PANI backbone.

Supplementary Fig. 51 IR spectrum of the gaseous products released during the popping reaction.

Safety Warnings:

The decomposition of HClO₄ produces Cl₂ gas. All reactions must be conducted in a fume hood equipped with a chlorine gas sensor.

Furthermore, the POP-C products from HClO₄-doped PANI with added free HClO₄ and from HCl-doped PANI with added free HClO₄ exhibit nearly identical CV curves to POP-C derived from standard PANI with 33% intrinsic free HClO₄ (Supplementary Fig. 52a). This confirms that all three precursors yield carbon with a high surface area and well-interconnected pore structure. These results lead to two key conclusions. First, the identity of the doped acid (HClO₄ or HCl) in the PANI salt is irrelevant to trigger the popping reaction. Second, the presence of free HClO₄ is the sole essential factor for initiating the process, irrespective of its origin. We also observed that PANI containing free HClO₄ resists complete drying, even after 72 h at 60 °C, as the acid forms a stable hydrate with water. Although this means water is a ubiquitous companion to free HClO₄ in reactive precursors, it does not undermine the central conclusion that free HClO₄ is the critical trigger. Finally, the comparable Raman spectra of all POP-C samples (Supplementary Fig. 52b) confirm that the carbon's fundamental structure, including hybridization, defect density, and level of disorder, is also determined by the presence of free HClO₄ and is independent of the initial doped anion.

Supplementary Fig. 52 Cyclic voltammetry curves of POP-C synthesized from PANI doped with HClO₄ containing intrinsic free HClO₄ (red line), PANI doped with HClO₄ containing exogenous free HClO₄ (purple line), and PANI doped with HCl containing exogenous free HClO₄ (green line).

7. Proposed mechanism for the formation of 2D carbon nanosheets from polyaniline by fast popping reaction

Based on our systematic investigation, we propose a multi-stage mechanism for the popping transformation of polyaniline into porous 2D carbon nanosheets (POP-C), as illustrated in Supplementary Fig. 53. The process is initiated by the coupled exothermic decomposition of free HClO₄ and the oxidation of the polymer backbone. This intense reaction rapidly releases a large volume of gases and substantial heat, that simultaneously exfoliates and carbonizes the precursor into the final 2D morphology.

Supplementary Fig. 53 (a) Schematic showing the structure of PANI (emeraldine salt, doped with HClO_4) containing free HClO_4 . (b) Schematic showing the decomposition of HClO_4 during heat accumulation. (c) Schematic showing the exothermic reaction, gas evolution and sheet formation.

Precursor requirements

The reaction requires a specific precursor composition: a PANI emeraldine salt (whose initial dopant anion, e.g., Cl^- or ClO_4^- , is inconsequential) containing a critical mass of free HClO_4 and water (possibly HClO_4 hydrate), as shown in Supplementary Fig. 53a. Our control experiments definitively establish that free HClO_4 is an irreplaceable component for initiating the ignition transformation.

Initial heating and HClO_4 decomposition

During the initial heating stage, the PANI matrix acts as a thermal insulator, promoting the localized accumulation of heat. This confined energy drives the dehydration and decomposition of the free HClO_4 hydrate, generating a suite of potent oxidants and radical species, including Cl_2O_7 (the anhydride of perchloric acid), ClO_2 , chlorinated oxygen radicals, and oxygen radicals^{36,37}, as shown in Supplementary Fig. 53b. The PANI backbone, known to function as a radical scavenger, can be used as additive to desensitize nanothermites³⁸⁻⁴⁰.

Exothermic reaction, gas evolution and sheet formation

Upon further heating, the confined oxidants and radicals trigger a violent, exothermic oxidation of the PANI backbone and the simultaneous decomposition of perchlorate species and chlorine oxides^{39,40}. These exothermic reactions generate substantial heat instantaneously, with the local temperature exceeding 1400 °C, as evidenced by the melting of a K-type thermocouple embedded in the precursor (Supplementary Fig. 3). Concurrently, these reactions release a large volume of gases. The rapid build-up of these gases creates a critical explosive pressure (Supplementary Fig. 53c). This pressure, combined with the intense thermal shock, forcibly exfoliates the PANI polymer layers in an event observed as the “popping”, ultimately yielding the characteristic large-area POP-C with 2D nanosheet structure. The system’s ability to accumulate heat, a function of precursor packing density, mass, and heating rate, is critical to reaching this popping threshold. If the PANI is HClO_4 -doped salt, the doped ClO_4^- also decompose. The near-complete absence of chlorine in the final POP-C, as confirmed by XPS (Supplementary Fig. 8 and Supplementary Table 2), indicates that chlorine is predominantly escape (in the form of Cl_2) from the carbon product. The violent gas evolution and rapid

expansion quench the reaction almost instantaneously, freezing the carbon structure. The combined effects of explosive exfoliation and in-situ templating by the gas bubbles yield the characteristic 2D nanosheet morphology with an open, interconnected pore network.

Reviewer #2:

The subject of this manuscript is interesting and aligns well with the scope of Nature Communications. The work is well-structured, and the experimental results are presented clearly. However, the authors should address the following points and revise the manuscript accordingly. Therefore I recommend major revisions before publication.

We appreciate the reviewer for his/her positive comments and kind advice, which are very helpful to us in improving the quality of our work. Careful revisions have now been made according to these questions and comments. The followings are our point-to-point responses.

Question 1.

Introduction – This section requires improvement.

The authors should discuss current trends in the ultrafast and spontaneous synthesis of carbon nanostructures. In this context, numerous recent studies have reported innovative approaches for producing amorphous carbon nanostructures (e.g., nanosheets, carbon dots, porous carbons), including those synthesized in flame-based environments via hypergolic reactions. These reactions, which employ an organic fuel and an oxidizer, are carried out under ambient conditions without the need for sophisticated apparatus. Interestingly, they share similarities in terms of reaction rapidity, popping and the audible sound generated during the process.

Please clarify how your synthesis differs from these methods (similarities etc). Relevant works on ultrafast carbon synthesis that should be cited include:

<https://pubs.acs.org/doi/full/10.1021/acsnano.4c10531>

<https://pubs.acs.org/doi/full/10.1021/acs.chemmater.4c02091>

<https://www.mdpi.com/1420-3049/26/6/1595>

Response: We thank the reviewer for this excellent suggestion and for directing us to the highly relevant literature on hypergolic and flame-based synthesis. We agree that these methods represent a significant and exciting trend in the rapid synthesis of carbon nanomaterials. We

have now thoroughly revised the Introduction section (Page 3-4 in the revised manuscript) to discuss these current trends, citing key references and added discussion of the similarities and distinctions of our work.

Page 3-4 in the revised manuscript:

Conventional synthesis of these carbon-based electrocatalysts, however, typically involves a multi-step process. This often includes energy-intensive, high-temperature pyrolysis (800-1200 °C) in a tube furnace under controlled atmospheres for several hours, followed by acid leaching and sometimes a second pyrolysis step to enhance active site density^{4,5}. Alternative methods, such as chemical vapor deposition, flash Joule heating, plasma carbonization, and ultrasonic spray pyrolysis, often have to be coupled with additional thermal treatment or require complex and costly equipment⁶⁻⁹. Recent innovative approaches have sought to circumvent these limitations by harnessing chemical energy for ultrafast transformations. Among these, hypergolic reactions, where separate fuel and oxidizer components spontaneously ignite upon contact, represent a powerful strategy for the rapid, ambient-condition synthesis of carbon nanostructures¹⁰⁻¹². In parallel, explosion- or shock-wave-assisted methods have also been developed, though they typically employ a top-down approach starting from bulk graphite, which can limit chemical tunability^{9,13-15}.

Building on the concept of chemical energy-driven synthesis, we introduce a rapid and energy-autonomous strategy using a polyaniline-HClO₄ composite as a single, self-contained precursor. The stored chemical energy is liberated through brief ignition triggered by mild heating (~120 °C), microwave irradiation, or mechanical stimulus at ambient temperature. This ignition drives the simultaneous exfoliation and carbonization of the polyaniline matrix, directly yielding ultrathin two-dimensional (2D) amorphous carbon nanosheets with a high surface area and an interconnected porous network. Unlike typical bipropellant hypergolic systems that require separate fuel and oxidizer, our monopropellant-inspired design integrates both roles into one stable solid. Upon triggering, the decomposition of HClO₄ generates intense local heat for instantaneous carbonization while simultaneously producing a violent gas release that acts

as a dynamic exfoliation force, forcibly separating the polymer into large-area nanosheets. The entire process is simple, conducted in a round-bottom flask without complex instrumentation or post-treatment, and completes within 2-7 min under ambient conditions. Moreover, in contrast to top-down approaches that limit chemical tunability, our bottom-up precursor synthesis enables facile molecular-level doping. We demonstrate this versatility by flexibly incorporating transition metal dopants (e.g., Fe, Co, Ni, Cu) during precursor preparation, creating single-atom-site tailored for high-performance electrocatalysis in the oxygen reduction, H₂O₂ production, and CO₂ reduction reactions.

10. Chalmpes, N. et al. Ultrahigh Surface Area Nanoporous Carbons Synthesized via Hypergolic and Activation Reactions for Enhanced CO₂ Capacity and Volumetric Energy Density. *ACS Nano* 18, 33491-33504 (2024).

11. Chalmpes, N., Tantis, I., Alsmail, A. W., Bourlinos, A. B. & Giannelis, E. P. Design, synthesis, and evaluation of noble metal nanoparticles and in situ-decorated carbon-supported nanoparticle electrocatalysts using hypergolic reactions. *Chem. Mater.* 36, 10616-10625 (2024).

12. Chalmpes, N. et al. Carbon nanostructures derived through hypergolic reaction of conductive polymers with fuming nitric acid at ambient conditions. *Molecules* 26, 1595 (2021).

Question 2.

Terminology – Consider replacing the term explosion with ignition, as the former may convey a negative connotation to readers.

Response: We thank the reviewer for this thoughtful suggestion. In accordance with this feedback, we have replaced the term “explosion” with the more appropriate “ignition” throughout the revised manuscript and the Supplementary Information to avoid any unintended negative connotations.

Question 3.

Reaction Yield and Atmosphere – What is the yield of the reaction? Possibly you face lower

yield than expected due to the ignition and the popping effect. Do you anticipate any differences if the reaction is conducted under an inert atmosphere? Is the reaction scalable?

Response: We sincerely thank the reviewer for raising this critical point regarding POP-C's production efficiency. The reviewer is correct that the violent gas release during the popping reaction can expel a portion of the product, potentially reducing the collectable yield. However, our initial reported yield of 10% (20 mg POP-C from 200 mg precursor) was misleading, as it was based on the total precursor mass. A typical reactive precursor is composed of 32% PANI emeraldine base (the carbon source), 15% doped HClO₄, 33% free HClO₄, and 20% water. Calculating the yield based solely on the PANI emeraldine base mass gives a carbon conversion efficiency of $20 \text{ mg} / (200 \text{ mg} \times 32\%) = 31\%$. To ensure reliability, yields were determined from at least three independent reactions. The average yield for a standard popping reaction (200 mg PANI precursor, heated at $15 \text{ }^\circ\text{C min}^{-1}$) was 28%. The primary cause of product loss is the physical expulsion of POP-C by the violent gas release during the popping reaction. Consequently, moderating the reaction intensity through precursor and reaction engineering is key to improving yield. For instance, reducing the PANI precursor amount yields a less violent reaction while still achieving a POP-C yield of 40%. Similarly, increasing the water content in the precursor decrease the reaction intensity, reducing the force of gas expulsion. This milder reaction minimizes product loss, resulting in a higher collected yield of 36%. We have revised the manuscript (Page 15) and the Supplementary Information (Page 30-34, 37, 40-41, 44-45, 49-50) accordingly to present this more accurate yield. We believe this corrected framework accurately reflects the carbon conversion efficiency of the popping process.

Page 15 in the revised manuscript:

Moreover, the yield of POP-C from the popping reaction ranges from 17% to 40% depending on reaction conditions, which is comparable to that of conventional pyrolysis (37%), as detailed in Supplementary Notes 4 and 5.

Page 30-34 in the Supplementary Information:

As detailed in the Experimental section and Supplementary Fig. 1, the PANI precursor for POP-

C was synthesized via interfacial polymerization of aniline. The oxidative polymerization occurs at the interface, forming polyaniline, which subsequently migrates into the aqueous phase (1.0 M HClO₄). The resulting product, confirmed to be emeraldine salt doped with HClO₄, was separated from the aqueous phase by centrifugation, washed with water, and air-dried at 60 °C for 22 h. The obtained PANI was in a wet, sticky state, indicating the likely presence of multiple components: the HClO₄ doped PANI emeraldine salt itself, residual free HClO₄ from the excess acid in the aqueous phase, and water. To quantitatively analyze these compositions, we developed a straightforward method based on acid-base titration, as illustrated in Supplementary Fig. 33.

Supplementary Fig. 33 Quantification protocol for the composition of the PANI precursor, including polyaniline base, doped HClO₄, free HClO₄, and water contents.

First, a known mass of the PANI sample was dispersed in an excess of 0.1 M KOH to ensure complete neutralization of all HClO₄ species. This includes both the doped HClO₄ (which forms a salt with the polyaniline backbone) and the free HClO₄ (residual acid from synthesis). The chemical reaction, in which the emeraldine salt is converted to the un-doped emeraldine base, is illustrated in Supplementary Fig. 34. Following this reaction, the PANI suspension turns from green to dark blue, visually confirming the formation of the emeraldine base (Supplementary Fig. 35a). After approximately 6 h, the PANI emeraldine base precipitates. The supernatant, containing the unreacted KOH, was then transferred to a beaker and titrated with

0.1 M HClO₄ using phenolphthalein (PhPh) as an indicator (Supplementary Fig. 35a). The amount of HClO₄ consumed in the initial neutralization was determined by calculating the difference between the initial KOH and the amount remaining after reaction, as quantified by the titration.

Supplementary Fig. 34 Schematic of the quantitative analysis for free and doped HClO₄ in PANI. (a) The PANI emeraldine salt is reacted with KOH. Titration of consumed KOH quantifies total HClO₄ (free and doped HClO₄). (b) The PANI emeraldine salt is dispersed in water. Titration of the aqueous phase quantifies free HClO₄. Doped HClO₄ is calculated by difference.

Supplementary Fig. 35 Quantitative analysis of HClO₄ in PANI. (a) Measurement of total HClO₄ content. (b) Measurement of free HClO₄ content. The doped HClO₄ content is the difference of total and free HClO₄ content.

Second, a known mass of the PANI sample was dispersed in water to dissolve the free HClO₄ (Supplementary Fig. 34). The suspension remained green, characteristic of the PANI emeraldine salt (Supplementary Fig. 35b). After approximately 6 h, the PANI emeraldine salt precipitated. The supernatant, which contained the dissolved free HClO₄, was transferred to a beaker and titrated with 0.1 M KOH using PhPh as an indicator (Supplementary Fig. 35b). The amount of free HClO₄ was directly quantified from the titration results. The mass of doped HClO₄ was subsequently calculated by subtracting the free HClO₄ content from the total HClO₄ content.

Third, a control sample of polyaniline without water and free HClO₄, designated PANI-D, was prepared. PANI-D was synthesized via the same interfacial polymerization of aniline as the standard PANI. The critical difference lies in the washing procedure: after separation from the aqueous phase, the polymer was washed with acetone to remove all water and free HClO₄, followed by air-drying at 60 °C for 22 h. The free HClO₄ originates from the residual HClO₄ solution trapped within the standard PANI matrix. Acetone washing effectively removes both this residual acid and the associated water. Consequently, PANI-D is obtained as a green powder, in contrast to the standard PANI, which contains water and free HClO₄ and presents as a black, sticky solid (Supplementary Fig. 1). The mass of polyaniline without doped HClO₄ (PANI base) was subsequently calculated by subtracting the doped HClO₄ content from the PANI-D content.

Finally, the water content can be obtained by subtracting the doped HClO₄ content, free HClO₄ content and PANI base content from the PANI content.

To investigate how the heating rate, precursor mass, water content, HClO₄ content, and the type of HClO₄ within the precursor influence the popping reaction, several PANI precursor were prepared and the PANI emeraldine base, doped HClO₄, free HClO₄ and water content was quantified based on the protocol in Supplementary Fig. 33. The quantitative compositional analysis via titration was performed at least three times for each sample to ensure reproducibility and to generate error bars.

Following the reviewer's insightful suggestion, we have systematic investigate the influence of PANI precursor amount, heating rate, water content, HClO₄ content on the popping reaction yield, which has been presented in the Supplementary Information (Page 37, 40-41, 44-45, 49-50). We found that milder reaction conditions minimize product loss, leading to a higher collected yield.

Page 37 in the Supplementary Information:

For the PANI used in this series of experiments (comprising 32% PANI base, 15% doped HClO₄, 33% free HClO₄, and 20% water), the yield decreased from 40% to 32% to 28% with

increasing precursor mass (Supplementary Fig. 38b). This trend is attributed to product loss during the violent reaction; a portion of the POP-C is expelled from the flask by the rapid gas generation and cannot be collected. Since the reaction intensity diminishes with lower precursor mass (as shown in Supplementary Fig. 36), the relative product loss is reduced, leading to a higher relative yield at lower masses.

Supplementary Fig. 38 Effect of PANI precursor mass on the popping reaction and resulting POP-C. (a) The reaction initiation temperature as a function of PANI mass. (b) The final yield of POP-C vs the initial PANI mass. (c) CV curves and (d) Raman spectra of POP-C synthesized from different PANI masses.

Page 40-41 in the Supplementary Information:

The product yield decreases from 34% to 32% to 26% with increasing heating rates (Supplementary Fig. 41b). This inverse relationship between yield and heating rate is attributed

to greater product loss during the more violent reactions; a larger portion of the POP-C is expelled from the flask by the rapid gas generation at higher heating rates and cannot be collected. This explanation is consistent with the observed increase in reaction intensity (i.e., brighter flame) at higher heating rates, as shown in Supplementary Fig. 40. The lowest triggering temperature observed was 86 °C, using 200 mg of PANI precursor and a heating rate of $\sim 30\text{ }^{\circ}\text{C min}^{-1}$. **However, combining a large precursor mass with a high heating rate can produce an excessively violent reaction. Therefore, this specific combination is not recommended due to safety concerns.**

Supplementary Fig. 41 Effect of heating rate on the popping reaction and resulting POP-C. (a) The dependence of reaction initiation temperature on PANI mass. (b) POP-C yield as a function of PANI mass. (c) CV curves and (d) Raman spectra of POP-C synthesized at different heating rates.

The product yield increases from 17% to 28% to 33% and finally to 36% with increasing water content (Supplementary Fig. 43b). This positive correlation is attributed to the less violent popping process observed at higher hydration levels. As the water content increases, the reaction intensity diminishes, resulting in less POP-C being expelled from the flask by rapid gas generation. The milder reaction thus minimizes product loss and leads to a higher collected yield.

A key consideration for scaling the popping reaction is managing its vigor. We have demonstrated that the reaction intensity can be effectively modulated by adjusting the water content of the precursor; higher water content reduces the intensity, albeit at the cost of a higher triggering temperature. *Therefore, precursors with elevated water content present a viable pathway for safer scale-up.*

Supplementary Fig. 43 Effect of water content in PANI precursor on the popping reaction and resulting POP-C. (a) The reaction initiation temperature and (b) final yield of POP-C as a function of water content. (c) CV curves and (d) Raman spectra of POP-C synthesized from PANI precursor containing different water

Page 49-50 in the Supplementary Information:

Correspondingly, the final POP-C yield decreases with increasing the free HClO_4 content (Supplementary Fig. 46b). This inverse relationship is attributed to greater product loss during the more violent reactions; a larger portion of POP-C is expelled from the flask by rapid gas generation and cannot be collected. As shown by the flame during the popping process in Supplementary Fig. 45, PANI containing 12% and 17% free HClO_4 content presents reddish flame, which PANI containing 33% and 47% free HClO_4 content present yellow flame, while the PANI containing 21% present orange color, indicating that the reaction temperature of the popping moment increase with the HClO_4 content.

Supplementary Fig. 46 Effect of HClO₄ content in PANI precursor on the popping reaction and resulting POP-C. (a) The reaction initiation temperature as a function of the free HClO₄ content. (b) The final yield of POP-C as a function of free HClO₄ content. (c) CV curves and (f) Raman spectra of POP-C synthesized from PANI precursors with varying free HClO₄ content.

The reviewer's suggestion to use an inert atmosphere is insightful. As detailed in Supplementary Note 5 (Page 63 in the Supplementary Information), we compared the popping reaction in air (yield: 17-40%) with conventional pyrolysis under N₂ (yield: ~37%). The comparable yields indicate that combustion in air is not a primary cause of product loss; the main loss mechanism is physical expulsion of material by the violent gas release. Therefore, performing the reaction in air yields an acceptable carbon conversion efficiency. Furthermore, the reaction's intrinsic chemistry makes an inert atmosphere functionally challenging to

achieve. The popping is triggered by the internal decomposition of HClO₄, which instantaneously releases a large volume of oxidizing gases (Cl₂, O₂) from within the precursor (Page 56 in the Supplementary Information). These gases rapidly create a highly oxidative local environment, making it practically difficult to maintain an inert blanket around the reacting solid.

Page 15 in the revised manuscript:

Moreover, the yield of POP-C from the popping reaction ranges from 17% to 40% depending on reaction conditions, which is comparable to that of conventional pyrolysis (37%), as detailed in Supplementary Notes 4 and 5.

Page 63 in the Supplementary Information:

The yield of POP-C from the popping reaction, detailed in Supplementary Note 4, ranges from 17% to 40%, depending on the precursor amount, hydration state, and other reaction conditions. For comparison, conventional pyrolysis of 200 mg of PANI-D (composed of 68% PANI base and 32% doped HClO₄, as quantified in Supplementary Table 4) at 800 °C for 2 h yields 50 mg of PANI-D800, corresponding to a carbon yield of 37%. Therefore, the yields from the rapid popping reaction and conventional pyrolysis are comparable.

Page 57 in the Supplementary Information:

During the reaction, chlorine can be detected in the fume hood by using a chlorine gas sensor, which is due to the decomposition of HClO₄. The decomposition of HClO₄ can also generate H₂O and O₂:

In the presence of polyaniline, additional gases are formed due to oxidation at high local temperatures. IR spectroscopy confirms the release of CO₂ (Supplementary Fig. 51), resulting from the strong oxidative carbonization of the PANI backbone.

Supplementary Fig. 51 IR spectrum of the gaseous products released during the popping reaction.

Safety Warnings:

The decomposition of HClO_4 produces Cl_2 gas. All reactions must be conducted in a fume hood equipped with a chlorine gas sensor.

The popping reaction presents significant potential for scalability due to its rapidity, low external energy requirement, and the commercial availability of its precursors. Its exothermic, self-propagating nature is inherently favorable for scale-up. A primary consideration for scalability is managing the reaction's vigor. We have demonstrated that this can be effectively modulated by adjusting the water content of the precursor; introducing water reduces the reaction intensity, albeit at the cost of a higher triggering temperature (Supplementary Information, Pages 42-44). Therefore, future scale-up efforts will focus on reactor design to safely manage the reaction pressure. The relevant discussion has been added to the revised manuscript:

Page 2 in the revised manuscript:

We rigorously validate that this energy-efficient method achieves a carbon conversion efficiency comparable to conventional pyrolysis and demonstrate that the reaction vigor is

precisely tunable via precursor water content, confirming its strong potential for safe scale-up

Page 15 in the revised manuscript:

Concurrently, vigorous outgassing promotes the formation of 2D carbon nanosheet. The popping process exhibits strong potential for scale-up due to its speed, low external energy demand, and tunable reaction intensity, which can be moderated via precursor water content, as detailed in Supplementary Notes 4.

Page 42-44 in Supplementary Information:

The PANI used to investigate the influence of water content was synthesized using 1.0 M HClO₄ as the aqueous phase, followed by water washing. To achieve different water contents, the drying time at 60 °C was systematically varied from 46 to 22 to 8 to 4 h, producing PANI with final water contents of 11%, 20%, 35%, and 58%, respectively. Precursors with these different water contents were then loaded into the reaction flask for the popping process. To ensure a constant mass of solid material (PANI base, doped HClO₄ and free HClO₄) across all experiments, the total PANI mass was adjusted to 180 mg (11% water), 200 mg (20% water), 250 mg (35% water), and 400 mg (58% water). As shown in Supplementary Video 7 and Fig. 42, the intensity of the popping reaction decreases as the water content in the PANI precursor increases. The popping process for PANI with 11% and 20% water content produces a significantly brighter flame than that observed for samples with 35% and 58% water, indicating a more violent reaction at lower hydration levels.

Supplementary Fig. 42 Synthesis of POP-C from PANI via rapid thermal popping with varying water contents: (a) 11%, (b) 20%, (c) 35%, and (d) 58%. The temperature displays alternates between the set-point and the actual thermocouple reading. The popping reaction initiates at the thermocouple reading (e.g., 163 °C in panel c, 184 °C in panel d), not at the displayed set-point value (e.g., 250 °C).

Both the reaction initiation temperature and the final POP-C yield are influenced by the water

content. As shown in Supplementary Fig. 43a, the initiation temperature increases from 114 °C to 124 °C to 164 °C and finally to 186 °C as the water content increases from 11% to 20% to 35% to 58%. During the heating process, water gradually evaporates from all types of PANI precursors before the popping reaction occurs. This evaporation is an endothermic process that cools the system. For PANI with higher water content, this heat loss is more severe, thereby hindering efficient heat accumulation. As discussed in the sections on the “Influence of PANI amount” and “Influence of the heating rate”, such heat accumulation is a critical factor for triggering the popping process. Consequently, precursors with higher water content require more time and a higher ultimate temperature to fully dehydrate and accumulate the substantial heat necessary to initiate the ignition reaction. Furthermore, high water content adversely affects the packing density, as the sticky, hydrated PANI adheres to the flask walls rather than consolidating into a dense mass. The product yield increases from 17% to 28% to 33% and finally to 36% with increasing water content (Supplementary Fig. 43b). This positive correlation is attributed to the less violent popping process observed at higher hydration levels. As the water content increases, the reaction intensity diminishes, resulting in less POP-C being expelled from the flask by rapid gas generation. The milder reaction thus minimizes product loss and leads to a higher collected yield.

A key consideration for scaling the popping reaction is managing its vigor. We have demonstrated that the reaction intensity can be effectively modulated by adjusting the water content of the precursor; higher water content reduces the intensity, albeit at the cost of a higher triggering temperature. *Therefore, precursors with elevated water content present a viable pathway for safer scale-up.*

Question 4.

Fuel Cell Performance – Fuel cell measurements have become a standard method for evaluating performance under realistic operating conditions. Consider including such data.

Response: We thank the reviewer for this insightful suggestion. We agree that device-level testing is crucial. Our half-cell measurements indicated that the Co-POP-C and Fe-POP-C

catalysts exhibit satisfactory ORR activity in alkaline media, but limited activity in acidic conditions typical of proton exchange membrane fuel cells. Therefore, to demonstrate their practical application in a relevant energy conversion device, we evaluated Co-POP-C in an aluminum-air battery, which operates in an alkaline electrolyte. As presented on Page 11 in the revised manuscript and Supplementary Fig. 27, the battery achieved an open-circuit voltage of 1.74 V, a peak power density of 160 mW cm⁻² and excellent discharge stability, thereby confirming the high practical performance of our catalyst.

Page 11 in the revised manuscript:

To evaluate the 4-electron ORR activity in a practical device, Co-POP-C was assembled into an Al-air battery as cathode catalyst. As shown in Supplementary Fig. 27, this battery demonstrated an open circuit voltage of 1.9 V and a peak power density of 160 mW cm⁻². During discharge at 20 mA cm⁻², the battery maintained a stable discharge voltage of 1.5 V and delivered a specific capacity of 950 mAh g⁻¹. This performance surpasses commercial Pt/C and is comparable to recently reported non-precious metal catalysts.

Page 25 in the Supplementary Information:

Supplementary Fig. 27 (a) Open circuit voltage of Al-air battery by using Co-POP-C as cathode catalyst. (a) Polarization (V-I) and power density curves recorded by LSV of Al-air battery by using Co-POP-C as cathode catalyst. The catalyst loading is 2 mg cm⁻². (b) Galvanostatic discharge curves at current density of 20 mA cm⁻².

Supplementary Table 3 Comparison of the Al-air battery performance with literature.

Catalyst	Open circuit voltage (V)	Power density (mW cm ⁻²)	Discharge voltage (V @ mA cm ⁻²)	Specific capacity (mAh g ⁻¹ @ mA cm ⁻²)	Energy density (Wh kg ⁻¹)	Ref.
Co-POP-C	1.9	160	1.5 @ 20	950 @ 20	1425	This work
CoNi@NCNTs/CC	1.68	151	1.28 @ 10	1029 @ 10	1317	29
SA-Fe-N _x -MPCS	1.55	130	1.35 @ 25	2204 @ 50	2975	30
FePc@HNSC	1.92	204.70	0.95 @ 100	1320 @ 100	1254	31
pFePc-TTF	1.93	218.34	1.6 @ 20	/	/	32
Fe ₂ N/CrN _x @NC	1.89	241.4	1.35 @ 100	2286 @ 100	3086	33
FePc/Eu ₂ O ₃	1.9	164.7	1.27 @ 100	/	/	34
Pt/C	1.6	136.1	1.20 @ 100	/	/	34

Question 5.

Language and Clarity – The manuscript would benefit from further refinement of language and sentence structure to improve clarity and readability.

Response: We thank the reviewer for this suggestion. We have thoroughly revised the manuscript to improve the language, sentence structure, and overall clarity. This includes correcting grammatical errors, refining awkward phrasing, and ensuring a more logical flow throughout the text.

Question 6.

XPS Fitting Accuracy – Verify the accuracy of the XPS fitting in the Supplementary Information. In particular, in Figures S17–S19, the FWHM of the deconvoluted peak attributed to oxidized nitrogen appears larger compared to the other components.

Response: We thank the reviewer for this insightful observation regarding the XPS peak fitting. We have carefully re-analyzed the C 1s, N 1s, and O 1s spectra (Supplementary Fig. 7, 17-19). Standardized full width at half maximum (FWHM) constraints were applied consistently across all component peaks, with the recognized exception of the π - π^* satellite peak in the C 1s spectrum, which intrinsically exhibits a wider FWHM. The revised deconvolutions are now more rigorous, improving the accuracy and reliability of our quantitative analysis. The updated

figures and corresponding data have been replaced in the revised manuscript (Page 6) Supplementary Information (Page 7 and 16-19).

Page 6 in the revised manuscript:

Fig. 2 Morphological and structural characterizations of POP-C. **a**, TEM image with a SAED pattern (inset). **b**, HRTEM image of the 2D carbon nanosheet. **c**, FFT filtered HRTEM image of the region outlined in **(b)**. **d**, Argon sorption isotherm and corresponding pore size distribution (inset). **e**, Neutron pair distribution function $G(r)$. **f**, XPS spectrum of the C 1s core level. **g**, XPS spectrum of N 1s core level. **h**, Carbon K-edge EELS. **i**, Carbon K-edge XANES spectrum.

Page 7 and 16-19 in the Supplementary Information:

Supplementary Fig. 8 XPS analysis of POP-C. (a) XPS survey showing predominant carbon, nitrogen, and oxygen signals with trace chromium from HClO_4 doping. High-resolution spectra of (b) C 1s, (c) N 1s, and (d) O 1s regions. The C 1s spectrum deconvolves into sp^2 C-C (284.5 eV), sp^3 C-C and C-N (285.7 eV), C-O (286.4 eV), C=O (287.5 eV), O-C=O (288.5 eV), as well as $\pi-\pi^*$ satellite peak (290.3 eV)⁵⁻⁹. N 1s components include: pyridinic N (398.8 eV), pyrrolic N (400.3 eV), graphilic N (401.2 eV) and oxidized N (402.8 eV)^{7,9,10}. O 1s peaks correspond to: C=O (531.2 eV), C-O (532.3 eV) and O-C=O (533.5 eV)^{5,6}.

Supplementary Fig. 17 XPS analysis of Fe-POP-C. (a) Survey spectrum. High-resolution spectra of (b) C 1s, (c) N 1s, (d) O 1s, and (e) Fe 2p regions.

Supplementary Fig. 18 XPS analysis of Co-POP-C. (a) Survey spectrum. High-resolution spectra of (b) C 1s, (c) N 1s, (d) O 1s, and (e) Co 2p regions.

Supplementary Fig. 19 XPS analysis of Ni-POP-C. (a) Survey spectrum. High-resolution spectra of (b) C 1s, (c) N 1s, (d) O 1s, and (e) Ni 2p regions.

Supplementary Fig. 20 XPS analysis of Cu-POP-C. (a) Survey spectrum. High-resolution spectra of (b) C 1s, (c) N 1s, (d) O 1s, and (e) Cu 2p regions.